# Acetylation of BMAL1 by TIP60 controls BRD4-P-TEFb recruitment to circadian promoters

Nikolai Petkau, Harun Budak[†], Xunlei Zhou[‡], Henrik Oster[§], Gregor Eichele*

Department of Genes and Behavior, Max Planck Institute for Biophysical Chemistry, Göttingen, Germany

*For correspondence:
gregor.eichele@mpibpc.mpg.de

Present address: [†]Department of Molecular Biology and Genetics, Science Faculty, Atatürk University, Erzurum, Turkey; [‡]Institute of Anatomy and Cell Biology, University of Heidelberg, Heidelberg, Germany; [§]Institute of Neurobiology,Center of Brain, Behavior and Metabolism, University of Lübeck, Lübeck, Germany

Competing interests: The authors declare that no competing interests exist.

**Abstract** Many physiological processes exhibit circadian rhythms driven by cellular clocks composed of interlinked activating and repressing elements. To investigate temporal regulation in this molecular oscillator, we combined mouse genetic approaches and analyses of interactions of key circadian proteins with each other and with clock gene promoters. We show that transcriptional activators control BRD4-PTEFb recruitment to *E-box*-containing circadian promoters. During the activating phase of the circadian cycle, the lysine acetyltransferase TIP60 acetylates the transcriptional activator BMAL1 leading to recruitment of BRD4 and the pause release factor P-TEFb, followed by productive elongation of circadian transcripts. We propose that the control of BRD4-P-TEFb recruitment is a novel temporal checkpoint in the circadian clock cycle.
DOI: https://doi.org/10.7554/eLife.43235.001

## Introduction

Circadian clocks are genetically encoded, self-sustained molecular oscillators that drive biological rhythms with a period close to 24 hr (*Partch et al., 2014*). They are found in most organisms and even in cultured tissues and cells (*Balsalobre et al., 1998*; *Brown et al., 2012*; *Yoo et al., 2004*). In mammals, these clocks are organized in a network controlled by a master pacemaker in the hypothalamic suprachiasmatic nucleus (SCN) (*Dibner et al., 2010*). Physiological processes such as energy metabolism, locomotion, hormone secretion, and cognitive functions are subject to circadian regulation (*Bass and Takahashi, 2010*; *Gerstner and Yin, 2010*; *Panda, 2016*). The influence of circadian rhythmicity is highlighted by the fact that in each tissue ~10% of the transcriptome is under the control of the circadian clock (*Zhang et al., 2014*) and that disruption of circadian rhythms is tied to the development of various diseases including metabolic disorders and cancer (*Bedrosian et al., 2016*; *Musiek and Holtzman, 2016*).

The molecular circadian oscillator consists of interlinked positive and negative feedback elements (*Reppert and Weaver, 2002*). Positive components are the heterodimers of CLOCK (circadian locomotor output cycles kaput) and BMAL1 (brain and muscle ARNT-like protein 1) that bind to *E-box* promoter elements of *Per1-3* (*Period*), *Cry1/2* (*Cryptochome*) and *Rev-ErbAα/β* (*Nr1d1/2*; nuclear receptor subfamily 1, group D, members 1 and 2) that encode repressor proteins PER, CRY and NR1D (*Takahashi, 2017*).

Genome-wide analysis of core circadian regulators revealed temporally restricted chromatin associations of circadian activators and repressors in the course of the circadian day (*Koike et al., 2012*; *Menet et al., 2012*; *Rey et al., 2011*). It is proposed that at the onset of the circadian cycle repressor CRY1 is part of the chromatin-bound CLOCK-BMAL1-complex located near transcriptionally silent RNA polymerase II (Pol II). Upon degradation of CRY1, coactivators such as CBP/p300 are recruited followed by *Per* and *Cry* transcription. During the repression phase of the circadian cycle, newly synthesized CRY2 and PER proteins associate with CLOCK-BMAL1, thereby shutting down

their own transcription (*Koike et al., 2012*). Recruitment of CRY to the CLOCK-BMAL1 heterodimer is facilitated by acetylation of BMAL1 at lysine 538 by the acetyltransferase activity attributed to CLOCK (*Doi et al., 2006*; *Hirayama et al., 2007*). BMAL1 acetylation peaks during the repression phase (*Hirayama et al., 2007*), although data by Nakakata and co-workers (*Nakahata et al., 2008*; *Nakahata et al., 2009*) suggest that the presence of acetylated BMAL1 extends into the activation phase.

An important issue that the above feedback mechanisms leave unresolved concerns the binding of CLOCK-BMAL1 heterodimers to *E-box* promoter elements that initiates Pol II-mediated transcription (*Koike et al., 2012*; *Stratmann et al., 2012*). Well-recognized, rate-limiting steps that regulate transcription are the recruitment of Pol II to promoters and Pol II release from promoter-proximal pause sites (*Jonkers and Lis, 2015*; *Liu et al., 2015*). Pause release of Pol II and the transition to productive elongation requires the activity of the P-TEFb (positive transcription elongation factor b) complex, composed of T-type cyclins and CDK9 (cyclin-dependent kinase 9) (*Jonkers and Lis, 2015*; *Liu et al., 2015*). P-TEFb is recruited to promoters through interaction with specific transcription factors and other proteins like BRD4 (bromodomain-containing protein 4) and SEC (super elongation complex) (*Jonkers and Lis, 2015*; *Liu et al., 2015*; *Luo et al., 2012*; *Shi and Vakoc, 2014*). After such recruitment, P-TEFb phosphorylates the carboxy-terminal domain (CTD) of Pol II at Ser2 and additionally phosphorylates the pausing factors NELF (negative elongation factor) and DSIF (DRB sensitivity inducing factor). Upon phosphorylation, NELF is evicted from Pol II and DSIF becomes a positive elongation factor. Pol II pausing serves as a checkpoint allowing for rapid and synchronous expression of genes (*Boettiger and Levine, 2009*; *Gilchrist et al., 2012*; *Lagha et al., 2013*; *Lin et al., 2011*; *Liu et al., 2015*).

BRD4 is a member of the bromo-domain and extra terminal domain (BET) protein family which bind through their bromo-domains to acetylated lysines of histones and of transcription factors (*Shi and Vakoc, 2014*). Bound BRD4 recruits P-TEFb to promoter proximal paused Pol II (*Jonkers and Lis, 2015*; *Liu et al., 2015*). Small molecule inhibitors that block BRD4 (e.g. JQ1) from binding to acetylated lysines prevent productive elongation (*Shi and Vakoc, 2014*). Thus BRD4 protein is a key player in the regulation of productive transcription elongation. Therefore, acetyltransferases that acetylate BRD4 binding partners would play an important role in regulating Pol II pause release. The lysine acetyltransferase TIP60 (60 kDa Tat-interactive protein) is recruited to active promoters (*Ravens et al., 2015*) and plays a role in early steps of transcription elongation in mammals and flies (*Kusch et al., 2014*; *Ravens et al., 2015*; *Shi et al., 2014*). TIP60 is an essential protein (*Hu et al., 2009*).

Here we provide evidence that TIP60 is a critical component of the circadian clock. During the activation phase of the circadian cycle, TIP60 acetylates BMAL1, which triggers recruitment of the BRD4-P-TEFb complex to *E-box*-containing circadian promoters leading to Pol II pause release and productive elongation of circadian transcripts. These findings suggest that pause release is a critical regulatory step in the transcription of circadian clock regulated genes and that TIP60 plays a central role in this process.

## Results

### Inhibition of CDK9 and BRD4 abolishes circadian oscillations

Synchronized fibroblasts stably expressing the clock-driven luciferase reporter *Bmal1-LUC* (*Nagoshi et al., 2004*) show sustained reporter expression with a period of ~26 hr. In the presence of flavopiridol (FP), a potent inhibitor of the CDK9 subunit of P-TEFb (*Rahl et al., 2010*), the amplitude of luciferase activity rhythms was suppressed in a dose dependent manner concomitant with an increased period length until, at high doses, the rhythm was fully lost (*Figure 1A*; *Figure 1—figure supplement 1A*). Treatment of fibroblasts with FP also strongly reduced endogenous mRNA expression levels of *Dbp* (D-site albumin promoter binding protein), *Per1*, and *Nr1d1* (*Figure 1—figure supplement 1C*) that are direct CLOCK-BMAL1 targets (*Stratmann et al., 2012*). These findings suggest that circadian clock gene regulation not only occurs at the initiation of transcription (*Le Martelot et al., 2012*) but also at the level of Pol II pause release and productive elongation. Triazolothienodiazepine (JQ1) is a potent and selective inhibitor of BRD4 (*Shi and Vakoc, 2014*). JQ1 treatment of *Bmal1-LUC* reporter fibroblasts dampened luciferase activity rhythms in a dose-

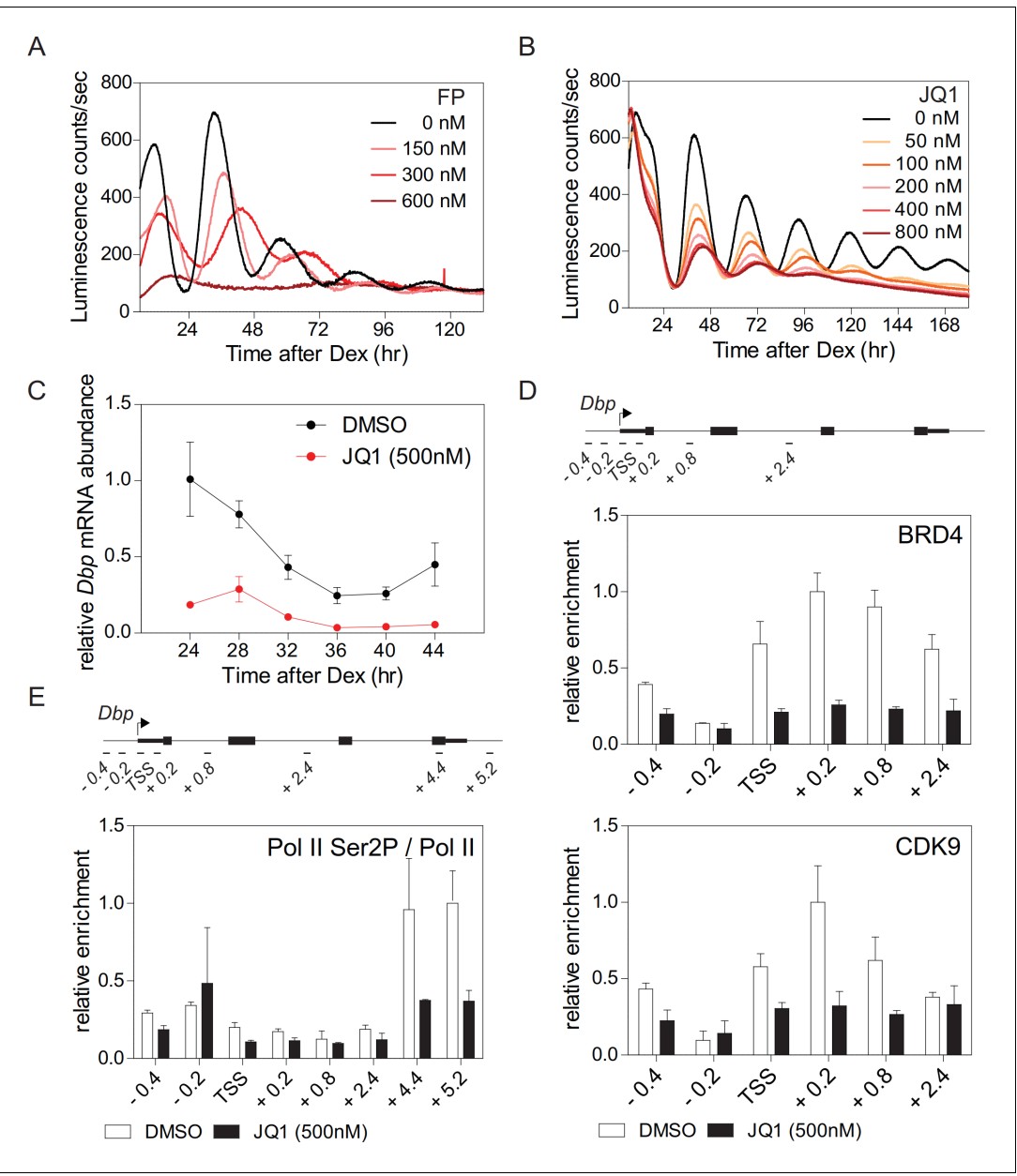

**Figure 1.** BRD4 controls clock gene expression. (**A and B**) Bioluminescence recordings of dexamethasone (Dex) synchronized *Bmal1-LUC* fibroblasts treated with increasing concentrations of flavoperidol (**A**) or JQ1 (**B**) (n = 4). (**C**) *Dbp* mRNA dampening in Dex-synchronized fibroblasts by 500 nM JQ1 (n = 3; two-way ANOVA, see *Supplementary file 1*). (**D and E**) ChIP analysis of BRD4 and CDK9 (**D**) and Ser2-phosphorylated Pol II (normalized to total Pol II) (**E**) for the *Dbp* gene carried out 24 hr after Dex synchronization of the cells (n = 3). Schematic diagrams show the genomic structure of the *Dbp* locus. Horizontal lines represent PCR-amplified genomic regions. All data are shown as mean ± SD. Note that *Dbp* has several additional intronic Pol II pausing sites (*Sobel et al., 2017*). For validation of ChIP-grade quality of antibodies see *Figure 1—figure supplement 2*.
DOI: https://doi.org/10.7554/eLife.43235.002

The following figure supplements are available for figure 1:

**Figure supplement 1.** BRD4 controls expression of *E-box*-controlled clock genes.
DOI: https://doi.org/10.7554/eLife.43235.003

**Figure supplement 2.** Validation of antibodies used for ChIP studies either deficient cells or antibody isotype controls were used to validate the ChIP-grade quality of the antisera.
DOI: https://doi.org/10.7554/eLife.43235.004

dependent manner and evoked a lengthening of the period (*Figure 1B*; *Figure 1—figure supplement 1B*). JQ1 also resulted in a strong dampening of the rhythm of endogenous *Dbp* expression in these cells (*Figure 1C*). In the presence of JQ1, enrichment of BRD4 and CDK9 at the transcription start site (TSS) of *Dbp* was markedly reduced (*Figure 1D*). Ser2 phosphorylation of Pol II, at the 3'-end of *Dbp*, was strongly diminished (*Figure 1E*; *Figure 1—figure supplement 1D*) suggesting that Pol II pause release had been suppressed (*Harlen and Churchman, 2017*). Reduced *Dbp* expression could be due to lower levels of BMAL1. To show that this is not the case, we constitutively expressed ectopic *CMV* promoter-driven *Bmal1-myc*. BRD4 inhibition by JQ1 strongly reduced endogenous expression levels of *Dbp* mRNA but had no effect on mRNA or protein levels of *Bmal1-myc* (*Figure 1—figure supplement 1E*). Furthermore, BRD4 inhibition neither affected endogenous mRNA nor protein levels of *Bmal1* (*Figure 1—figure supplement 1F*) in *Bmal1-LUC* fibroblasts at 24 hr after Dex synchronization. Finally, the binding of BMAL1 and of acetylated BMAL1 (AcBMAL1) to the *Dbp* promoter was not affected by BRD4 inhibition in these cells (*Figure 1—figure supplement 1G*). Together, our JQ1 inhibitor studies suggest that BRD4-P-TEFb is recruited to the *E-box*-containing clock-controlled gene *Dbp* which subsequently allows Pol II pause release and transcription elongation. Thus, for this circadian clock-controlled gene, Pol II pause release may serve as an important regulatory step of its transcription.

## Acetylation of BMAL1 at lysine 538 is required to initiate circadian productive transcription elongation

BRD4 binds to acetylated lysines of transcription factors/histones at the promoter of paused genes (*Shi and Vakoc, 2014*). The observed strong effect of JQ1 on the cellular circadian rhythm raises the possibility that among BRD4 targets there are core clock components, such as BMAL1, which is acetylated at Lys538 (*Hirayama et al., 2007*) and could therefore provide a binding site for BRD4. We mutated this particular lysine to an arginine (BMAL1$^{K538R}$) using the CRISPR/Cas9 technique (*Figure 2—figure supplement 1A–1C*). BMAL1$^{K538R}$ protein was as stable as wildtype BMAL1 (*Figure 5—figure supplement 1A*), and subcellular localization and phosphorylation status of both forms were similar (*Hirayama et al., 2007*). We also found an unchanged enrichment of both BMAL1 variants at the promoters of the *Dbp*, *Per1*, and *Nr1d1* genes (*Figure 2—figure supplement 1C*). Next, the interaction of BRD4 and CDK9 with BMAL1 was examined in control fibroblasts and fibroblasts containing engineered BMAL1$^{K538R}$. Immunoprecipitation experiments showed that in BMAL1$^{K538R}$ expressing cells this interaction was markedly reduced (*Figure 2A*). When wildtype cells were JQ1-treated, this interaction was substantially dampened (*Figure 1—figure supplement 1H*). ChIP experiments demonstrated that BMAL1$^{K538R}$ mutant cells showed a diminished recruitment of BRD4 and of the P-TEFb subunit CDK9 to the TSS of *Dbp* (*Figure 2B*), *Per1*, and *Nr1d1* (*Figure 2—figure supplement 1E*). In addition, enrichment of Ser2-phosphorylated Pol II was markedly reduced (*Figure 2B*; *Figure 2—figure supplement 1E*; *Figure 2—figure supplement 2*). Since both, control fibroblasts and BMAL1$^{K538R}$ mutant cells stably expressed a clock-driven luciferase reporter, we assessed the effect of the K538R mutation on luminescence rhythms. In BMAL1$^{K538R}$ cells virtually no rhythmic luciferase reporter expression was recorded for any of the three clones (*Figure 2C*). In addition, in these cells there was a pronounced reduction of peak expression of endogenous *Dbp* (*Figure 2D*), *Per1*, and *Nr1d1* mRNAs (*Figure 2—figure supplement 1F*). It is possible that BMAL1 acetylation could additionally promote transcription initiation. However, the occupancy of the general transcription factor TFIIEα (part of the Pol II transcription initiation complex; *Sainsbury et al., 2015*) at the TSS of *Dbp*, *Per1*, and *Nr1d1* genes was not affected in mutant cells (*Figure 2—figure supplement 1G*). This favors the idea that transcription elongation, and not initiation, is the process that is primarily regulated by acetylation of BMAL1. Our data support a mechanism in which acetylated BMAL1 recruits BRD4-P and TEFb to *E-box*-containing circadian clock genes which then results in a release of Pol II from its paused state, thereby allowing productive elongation.

## TIP60 is essential for a functional circadian clock in the mouse

BMAL1 acetylation is carried out by CLOCK (*Hirayama et al., 2007*). However, we found that in CLOCK-deficient fibroblasts BMAL1 was acetylated at Lys538 to a similar extent as observed in wildtype cells (*Figure 3A*). CLOCK$^{mutA}$ fails to acetylate histones H3 and H4 (*Doi et al., 2006*) and shows greatly diminished activity in the acetylation of BMAL1 (*Hirayama et al., 2007*). We found that both

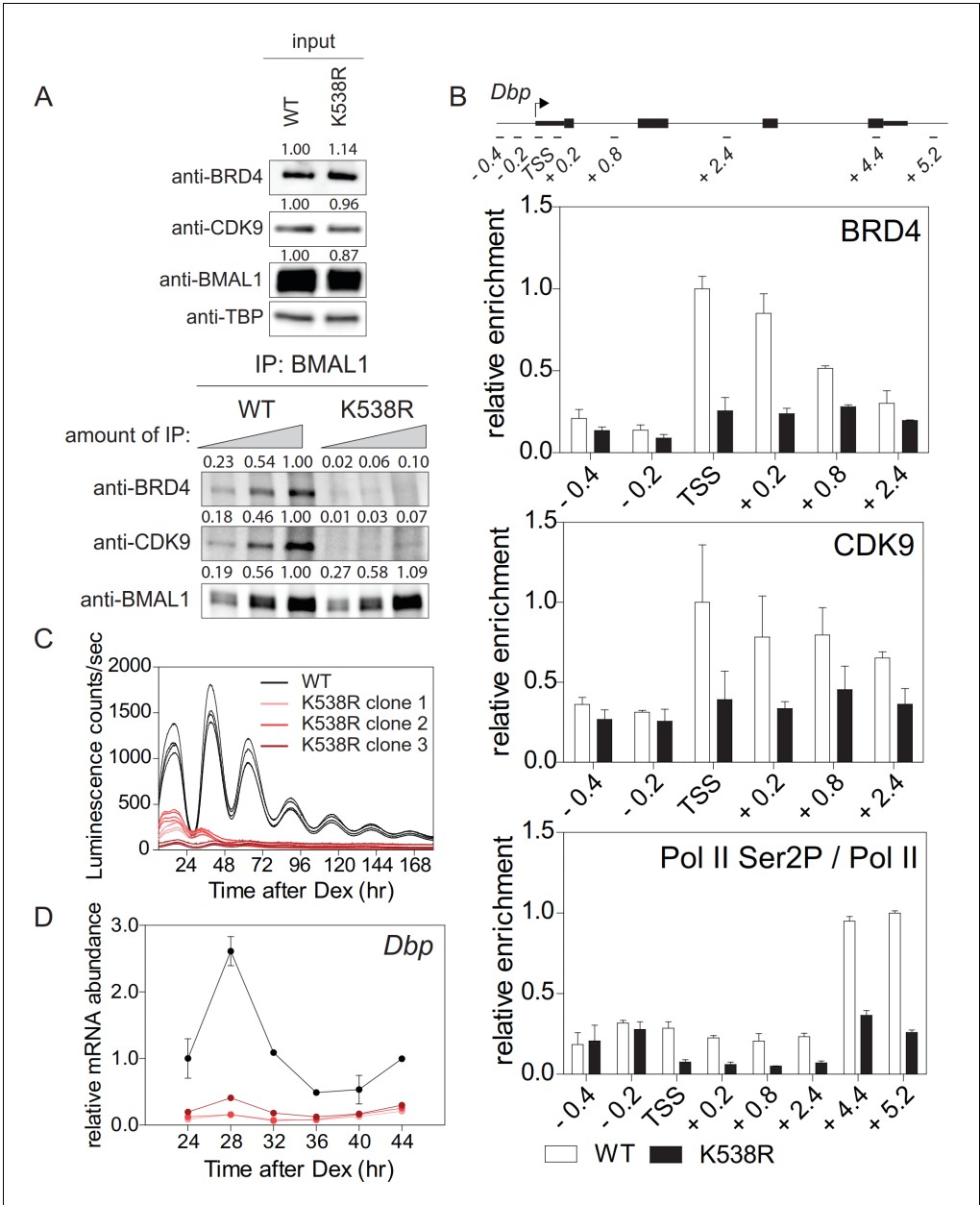

**Figure 2.** Lys538 acetylation of BMAL1 Is essential for transcription elongation. (**A**) Interaction of BMAL1 or BMAL1[K538R] with BRD4 and CDK9 as shown by immunoblotting of nuclear extracts from synchronized fibroblast 24 hr after Dex treatment (two-fold dilutions). TBP (TATA-binding protein) was used as loading control. Numerical values represent intensities of chemiluminescence signals of individual bands, normalized to wildtype and loading control for input samples. (**B**) ChIP analysis of BRD4, CDK9 and Ser2P-Pol II (normalized to total Pol II) binding to the *Dbp* gene in wildtype and BMAL1[K538R] synchronized fibroblasts as shown in (**A**) (n = 3). (**C**) *Bmal1-LUC* bioluminescence tracings of synchronized wildtype fibroblasts and of three independent BMAL1[K538R] clones (n = 4). (**D**) *Dbp* mRNA expression analysis of wildtype or BMAL1[K538R] fibroblasts taken from the experiment shown in (**C**) (n = 3, two-way ANOVA, see ***Supplementary file 1***). All data are shown as mean ± SD.

DOI: https://doi.org/10.7554/eLife.43235.005

The following figure supplements are available for figure 2:

**Figure supplement 1.** CRISPR/Cas9-mediated generation and characterization of BMAL1[K538R] mutant cells.

DOI: https://doi.org/10.7554/eLife.43235.006

**Figure supplement 2.** Characterization of BMAL1[K538R] mutant cells Pol II and Ser2P-Pol II ChIP analysis of Dex synchronized wildtype or BMAL1[K538R] fibroblasts (n = 3).

*Figure 2 continued on next page*

*Figure 2 continued*

DOI: https://doi.org/10.7554/eLife.43235.007

CLOCK variants showed a minimal increase of BMAL1 acetylation compared to a control (*Figure 3B*). Importantly, BMAL1 acetylation levels in the presence of CLOCK or CLOCK[mutA] were indistinguishable. Together, these data suggest the existence of alternative acetyltransferases that catalyze BMAL1 acetylation. A candidate is TIP60 that co-purifies with CLOCK-BMAL1 (*Doi et al., 2006*) and acetylates a broad range of transcription factors (*Judes et al., 2015*). Addition of TIP60, but not enzymatically inactive TIP60[C369A;E403Q] (*Yang et al., 2012*), strongly enhanced BMAL1 acetylation (*Figure 3B*).

This unexpected discovery of a new potential regulator of the circadian clock prompted us to generate conditional TIP60 mice to assess the in-vivo function of this enzyme and provide fibroblasts that we then used to study the molecular clock in vitro. Tip60[fl/-] mice carrying one conditional and one deleted allele of *Tip60* (*Figure 4—figure supplement 1A and B*) were mated to *Syt10-CRE* driver mice. This driver is strongly active in the postmitotic neurons of the SCN (*Husse et al., 2011*) circumventing cell lethality associated with TIP60-deficiency (*Hu et al., 2009*). All offspring carrying two copies of the knock-in *Cre* allele (*Syt10[Cre/Cre] Tip60[fl/-]*) and half of the mice carrying one copy of the knock-in *Cre* allele (*Syt10[Cre/+]Tip60* fl/-) became immediately arrhythmic upon release into

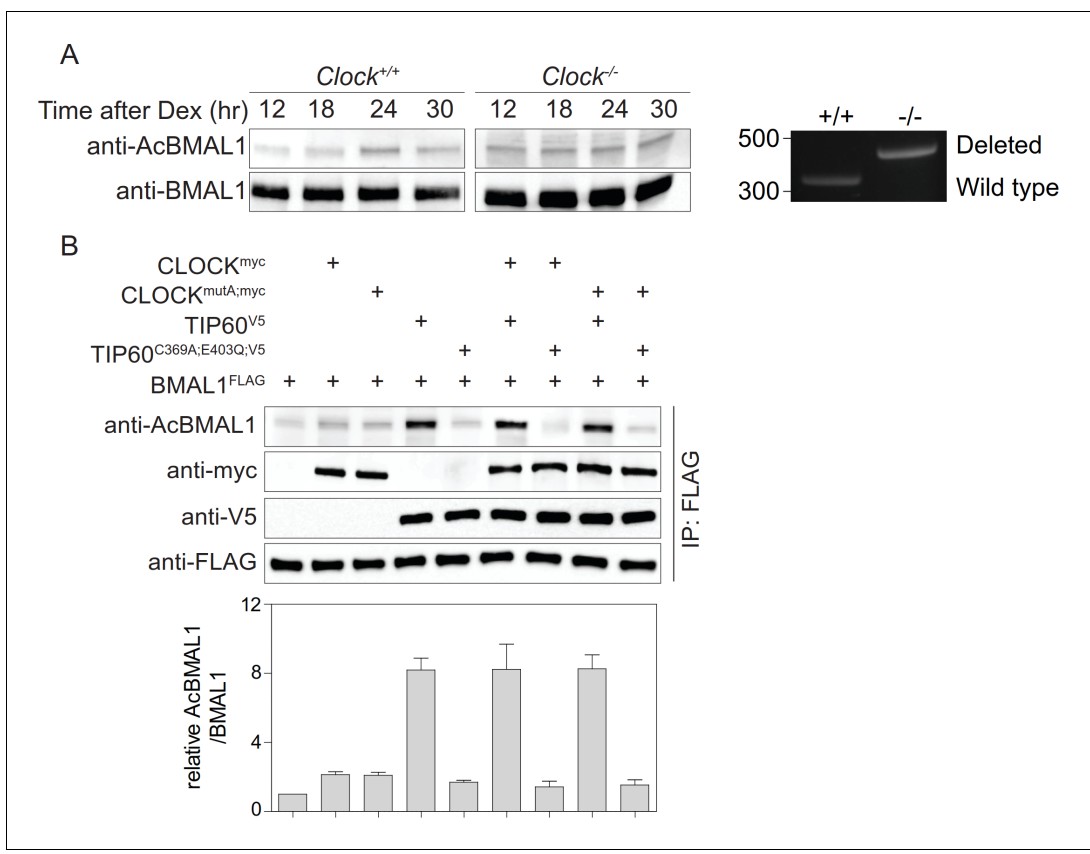

**Figure 3.** Acetylation of BMAL1 in CLOCK-deficient cells. (**A**) BMAL1 IP analysis of control and *Clock*-deficient MEF nuclear extracts over a 24 hr time course. For both genotypes, equivalent amounts of BMAL1 were included as shown in the anti-BMAL1 probed western blot. Lys538 acetylation of BMAL1 was detected using an anti-AcBMAL1 antibody (left). Wildtype and mutant cells were PCR-genotyped to confirm their genotype (right) (*Debruyne et al., 2006*). (**B**) BMAL1[FLAG], CLOCK[myc], CLOCK[mutA;myc], TIP60[V5], and TIP60[C369A;E403Q;V5] were transiently overexpressed in HEK293T cells in the combinations indicated. Lysates were subjected to IPs and immunoblotted with antibodies indicated (n = 3). Data are shown as mean ± SD.

DOI: https://doi.org/10.7554/eLife.43235.008

constant darkness (DD) (*Figure 4A and B*). The rest of the *Syt10*$^{Cre/+}$ *Tip60* fl/- mice showed either longer (29%) or normal (21%) free-running periods. Such a *Cre* dosage dependence was also observed with *Syt10-CRE*-mediated deletion of *Bmal1*$^{fl/fl}$ and, therefore, characterizes this particular driver (*Husse et al., 2011*). Immunohistochemical staining for TIP60 in the SCN showed nuclear signal, which was absent in arrhythmic *Syt10*$^{Cre/+}$ *Tip60* fl/- mice (*Figure 4C*). Moreover, circadian expression patterns of *Per1*, *Dbp*, and *Bmal1* in the SCN were severely dampened (*Figure 4D*) in line with the behavioral arrhythmicity observed in these animals. Nuclear staining of SCN sections from *Syt10*$^{Cre/+}$ *Tip60*$^{+/+}$ and arrhythmic *Syt10*$^{Cre/+}$ *Tip60* fl/- mice showed no obvious changes in SCN morphology or in the number of SCN cells (control: 827 ± 60 cells per section; mutant: 850 ± 68 cells per section). In addition, no TUNEL signal, indicating apoptotic cells, could be

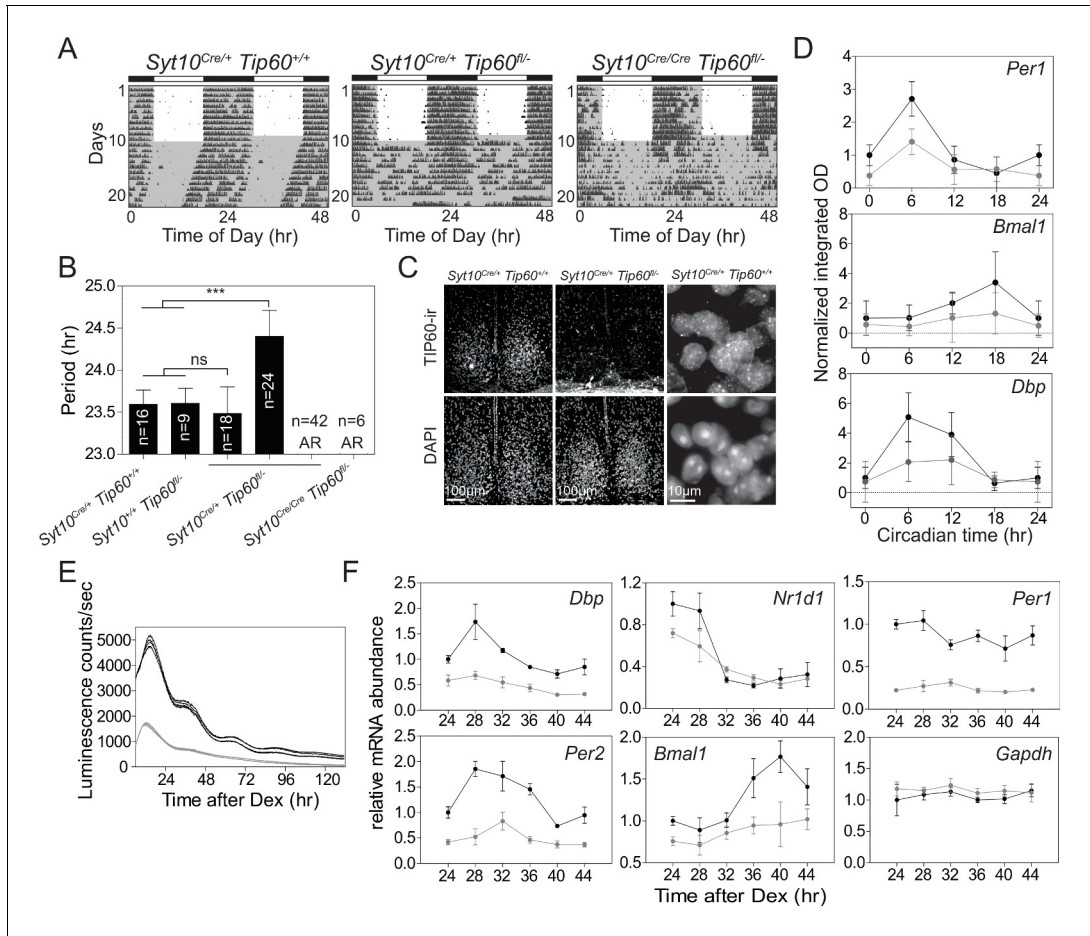

**Figure 4.** TIP60-deficiency evokes a circadian phenotype in mice and disrupts rhythmic clock gene expression in the SCN and in MEFs. (**A**) Double-plotted actograms of control (*Syt10*$^{Cre/+}$*Tip60*$^{+/+}$) and mutant (*Syt10*$^{Cre/+}$*Tip60* fl/-, *Syt10*$^{Cre/Cre}$ *Tip60*$^{fl/-}$) mice under 12 hr:12 hr light-dark and constant darkness conditions. Gray shadings indicate dark phases. (**B**) Free-running periods determined by χ$^2$ periodogram analysis. AR: arrhythmic (***p<0.001, one-way ANOVA with Bonferroni post-test). (**C**) TIP60-immunoreactivity (ir) in the SCN of control (*Syt10*$^{Cre/+}$*Tip60*$^{+/+}$) and arrhythmic mutant (*Syt10*$^{Cre/+}$*Tip60* fl/-) mice. (**D**) Densitometric quantification of clock gene mRNA expression from radioactive in situ hybridization analysis of control (*Syt10*$^{Cre/+}$*Tip60*$^{+/+}$; black) and mutant (*Syt10*$^{Cre/+}$*Tip60* fl/-; gray) SCN sections (n = 3; two-way ANOVA, see *Supplementary file 1*). (**E**) Bioluminescence recordings of synchronized *Tip60*$^{fl/-}$; *Bmal1-LUC* MEFs transduced with AdGFP (black) or AdCre (gray) (n = 4). (**F**) mRNA expression analysis of Dex synchronized *Tip60*$^{fl/-}$ MEFs transduced with either AdGFP (black) or AdCre (gray) (n = 3; two-way ANOVA, see *Supplementary file 1*). All data are shown as mean ± SD.

DOI: https://doi.org/10.7554/eLife.43235.009

The following figure supplement is available for figure 4:

**Figure supplement 1.** Generation and validation of the *Tip60*-deficient mice and MEFs derived from such animals.

DOI: https://doi.org/10.7554/eLife.43235.010

detected in the SCN of TIP60-deficient animals (*Figure 4—figure supplement 1C*). Together, these experiments indicate that SCN cells were fully viable after deletion of *Tip60*.

The essential circadian function of TIP60 was also seen in MEFs. Immortalized MEFs were prepared from *Tip60^{fl/-}* mouse embryos and were transduced with a *Bmal1-LUC* reporter. Confluent, that is non-dividing, MEFs were infected with AdCre to delete TIP60. Cell confluency is sufficient to induce cell cycle quiescence (*Hayes et al., 2005*), thus bypassing the role of TIP60 in cell cycle regulation (*Sykes et al., 2006*; *Tang et al., 2006*). Deletion resulted in a major dampening of the rhythmic expression of the luciferase reporter and a significantly reduced expression of endogenous clock genes (*Figure 4E and F*), but not of several housekeeping genes (*Figure 4F*; *Figure 4—figure supplement 1D*). The deletion of TIP60 in confluent MEFs did not affect cell viability (*Figure 4—figure supplement 1E*) nor did adenoviral transduction itself alter the circadian rhythm of a luciferase reporter (*Figure 4—figure supplement 1F*). Thus, abrogation of TIP60 function has major effects on locomotor activity and the rhythmic expression of clock genes in the SCN in vivo and in postmitotic MEFs.

## TIP60 acetylates BMAL1 at lysine 538

The severe circadian phenotype in the mouse, the strong changes of clock gene expression in TIP60-deficient SCN and MEFs, and co-transfection experiments (*Figure 3B*) all suggest that BMAL1 could be a direct substrate of TIP60. Co-transfection experiments showed that Lys538 of BMAL1 was acetylated by TIP60 in a dose-dependent manner and that acetyltransferase-deficient TIP60^{C369A;E403Q} (*Yang et al., 2012*) led to baseline acetylation levels of BMAL1 similar to mock transfected cells (*Figure 3B*; *Figure 5A*). Lys538 appeared to be the predominant lysine acetylated by TIP60 as the signal obtained with an anti-pan-AcK antibody was strongly diminished in cells co-transfected with BMAL1^{K538R} and TIP60 (*Figure 5A*). TIP60 co-precipitated with wildtype and BMAL1^{K538R} suggesting an interaction between the two proteins (*Figure 5A*; *Figure 5—figure supplement 1A*). Acetylation of BMAL1 did not influence this interaction. Both, BMAL1 and TIP60^{V5} proteins bound to the *E-boxes* of the *Dbp* gene (*Figure 5—figure supplement 1B*). Recombinant TIP60^{GST} acetylated recombinant BMAL1^{GST} at Lys538 (*Figure 5B*) and, in TIP60-deficient MEFs, Lys538 acetylation of endogenous BMAL1 was markedly reduced (*Figure 5C*). When endogenous TIP60 was replaced by TIP60^{C369A;E403Q} in the *Bmal1-LUC* reporter cell line, luciferase rhythms and mRNA rhythms of endogenous *Dbp* were abolished (*Figure 5D and E*). Collectively, these data provide strong evidence that BMAL1 is a substrate of TIP60.

## TIP60 controls productive elongation of circadian transcripts

So far, we have shown that TIP60 acetylates BMAL1 at Lys538 and that abrogation of TIP60 function has major effects on the rhythmic expression of clock genes (*Figure 4*). BMAL1^{K538R} failed to interact with BRD4 and P-TEFb (*Figure 2A and B*). Furthermore, Pol II Ser2 phosphorylation, an indicator of productive elongation, was markedly reduced in BMAL1^{K538R} cells (*Figure 2B*, *Figure 2—figure supplement 1E*; *Figure 2—figure supplement 2*). It follows that deletion of TIP60 should abrogate BRD4-P-TEFb recruitment and Pol II pause release. Indeed, in confluent TIP60-deficient Dex synchronized fibroblasts, BMAL1 was hypoacetylated (*Figures 5C* and *6A*) and there was a major loss of interaction between BMAL1 and BRD4 (*Figure 6A and B*) as well as between BMAL1 and CDK9 (*Figure 6C*). ChIP experiments showed that enrichment of BRD4 and CDK9 at the TSS of *Dbp*, *Per1*, and *Nr1d1* genes was strongly reduced and Ser2-phosphorylated Pol II occupancy was diminished (*Figure 6D*; *Figure 6—figure supplement 1A and B*).

Above we provided evidence that acetylation of BMAL1 plays a key role in transcription elongation rather than initiation (*Figure 2—figure supplement 1G*). Since TIP60 acetylates BMAL1, TIP60 deletion should not affect transcription initiation, which was indeed the case. Deletion of *Tip60* did not affect TFIIEα recruitment to the TSS of *Dbp*, *Per1*, and *Nr1d1* genes (*Figure 6—figure supplement 1C*).

## Rhythmicity of productive elongation of circadian transcripts

Collectively, our experiments provide evidence that TIP60-mediated acetylation of BMAL1 is an essential element of the positive limb of the circadian clock oscillator. TIP60-mediated BMAL1 acetylation leads to the recruitment of BRD4 and of the pause release factor P-TEFb to clock gene TSSs.

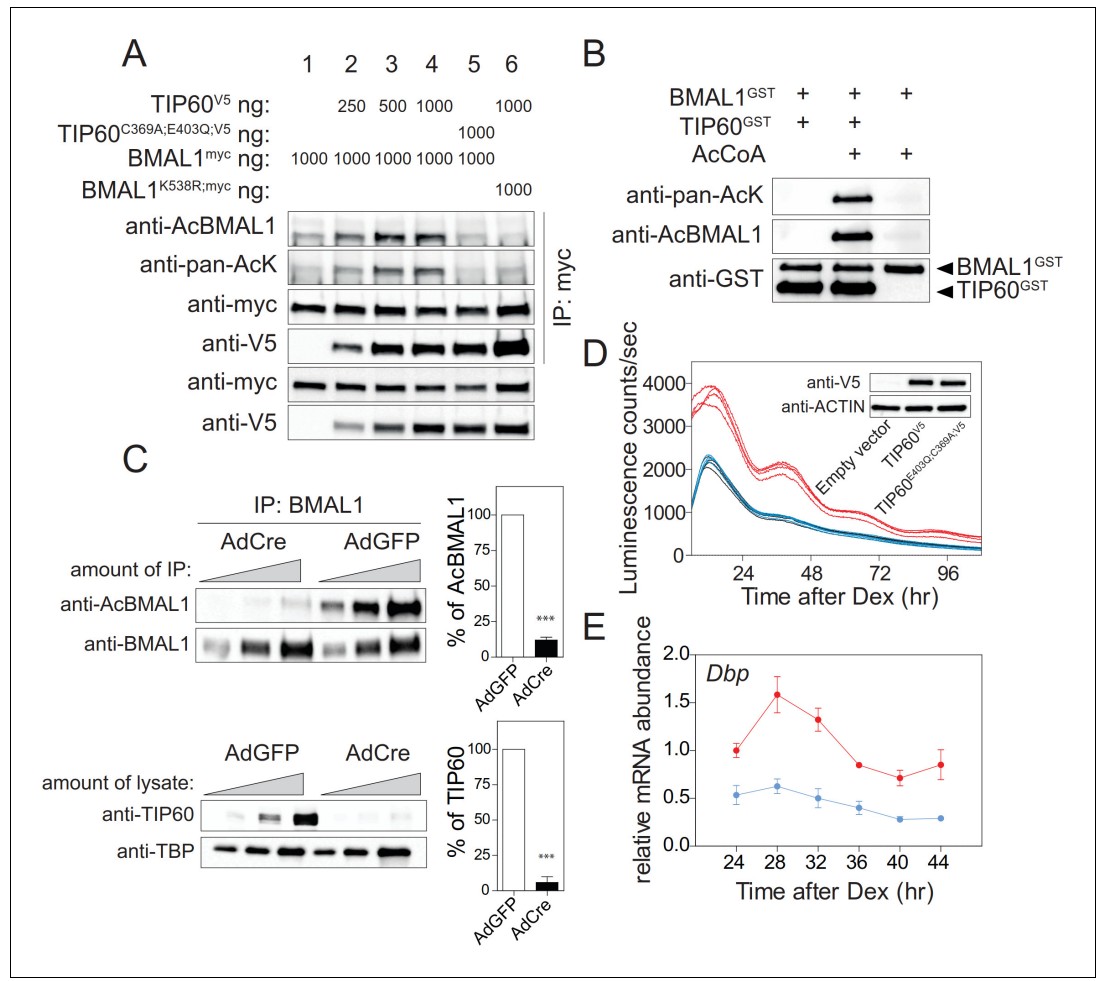

**Figure 5.** TIP60 acetylates BMAL1. (**A**) BMAL1$^{myc}$, BMAL1$^{K538R}$, TIP60$^{V5}$, and TIP60$^{C369A;E403Q;V5}$ were transiently overexpressed in HEK293T cells in the combinations indicated on top. Lysates were subjected to IPs using the antibodies indicated. (**B**) Recombinant BMAL1$^{GST}$, TIP60$^{GST}$ and acetyl-CoA were incubated and the presence of acetylated BMAL1 was detected by immunoblotting using either a pan-acetyl or a Lys538-specific AcBMAL1 antibody. (**C**) IPs (top) and nuclear extracts (bottom) from AdGFP (control) and AdCre (mutant) transduced unsynchronized *Tip60$^{fl/-}$* MEF (two-fold dilutions) were immunoblotted with antibodies indicated and signal intensities were quantified and normalized. Data are shown as mean of all relative values ± SD (n = 3). (**D**) *Tip60$^{fl/-}$; Bmal1-LUC* MEFs stably expressing TIP60$^{V5}$ (red), TIP60$^{C369A;E403Q;V5}$ (blue) or empty vector (black, coinciding with the blue tracing) were transduced with AdCre and bioluminescence was recorded (n = 4). (**E**) *Dbp* expression profiles in TIP60$^{V5}$ and TIP60$^{C369A;E403Q;V5}$ cells shown in (**D**). (n = 3; two-way ANOVA, see **Supplementary file 1**).
DOI: https://doi.org/10.7554/eLife.43235.011

The following figure supplement is available for figure 5:

**Figure supplement 1.** TIP60 interacts with BMAL1 on chromatin.
DOI: https://doi.org/10.7554/eLife.43235.012

This, in turn, allows productive elongation of CLOCK-BMAL1-controlled genes. This positive limb of the circadian cycle is counteracted by a negative feedback. This balance between positive and negative regulation, characteristic for the circadian cycle, should also be seen in temporal occupancy profiles of pause release factors at the promoters of *E-box*-controlled clock genes. Approximately 24 hr after synchronization of the cells, we observed the strongest enrichment of BMAL1 and of acetylated BMAL1 at the promoters of *Dbp*, *Per1*, and *Nr1d1* genes (**Figure 7A**, green). The concentration of acetylated BMAL1 in nuclear extracts also showed its maximum by that time (**Figure 7B**). In parallel, the occupancy of BRD4 peaked at target gene promoters (**Figure 7A**) and Ser2 phosphorylation of Pol II showed a maximum (**Figure 7A**; **Figure 7—figure supplement 1**). Consequently, the

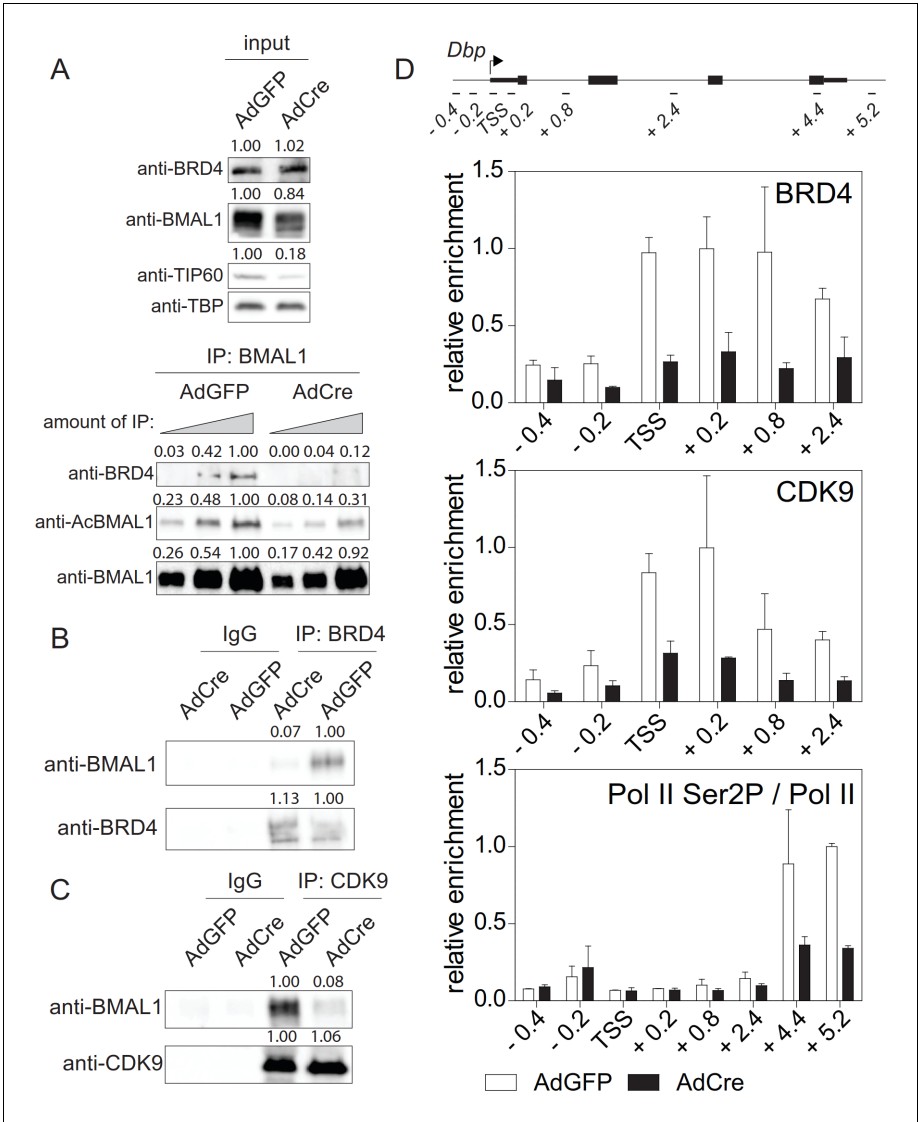

**Figure 6.** TIP60 controls productive elongation. (A) Interaction of BMAL1 with BRD4 in *Tip60^{fl/-}* MEFs transduced with either AdGFP (control) or AdCre (mutant). Immunoblots show results with increasing equivalents of IPs (two-fold dilutions) from nuclear extracts isolated from cells 24 hr after Dex synchronization. (B) BRD4 or (C) CDK9 IPs from AdGFP or AdCre transduced *Tip60^{fl/-}* MEF nuclear extracts were immunoblotted with antibodies indicated. Analyses were carried out 24 hr after Dex synchronization. (D) ChIP analysis for BRD4, CDK9 and Ser2P-Pol II (normalized to total Pol II) in the promoter or the 3'-end of *Dbp* for the cells shown in (A). Data are shown as mean ± SD (n = 3). Numerical values represent intensities of chemiluminescence signals of individual bands, normalized to wildtype and loading control for input samples.
DOI: https://doi.org/10.7554/eLife.43235.013

The following figure supplement is available for figure 6:

**Figure supplement 1.** TIP60 controls productive elongation of circadian genes.
DOI: https://doi.org/10.7554/eLife.43235.014

abundance of *Dbp*, *Per1*, and *Nr1d1* mRNA peaked between 24 and 28 hr (*Figure 7A*). Taken together, our data show that TIP60-dependent BRD4 recruitment, Pol II pause release, and productive elongation are precisely timed over the circadian cycle and in this way exert a temporal control over the circadian clock oscillator.

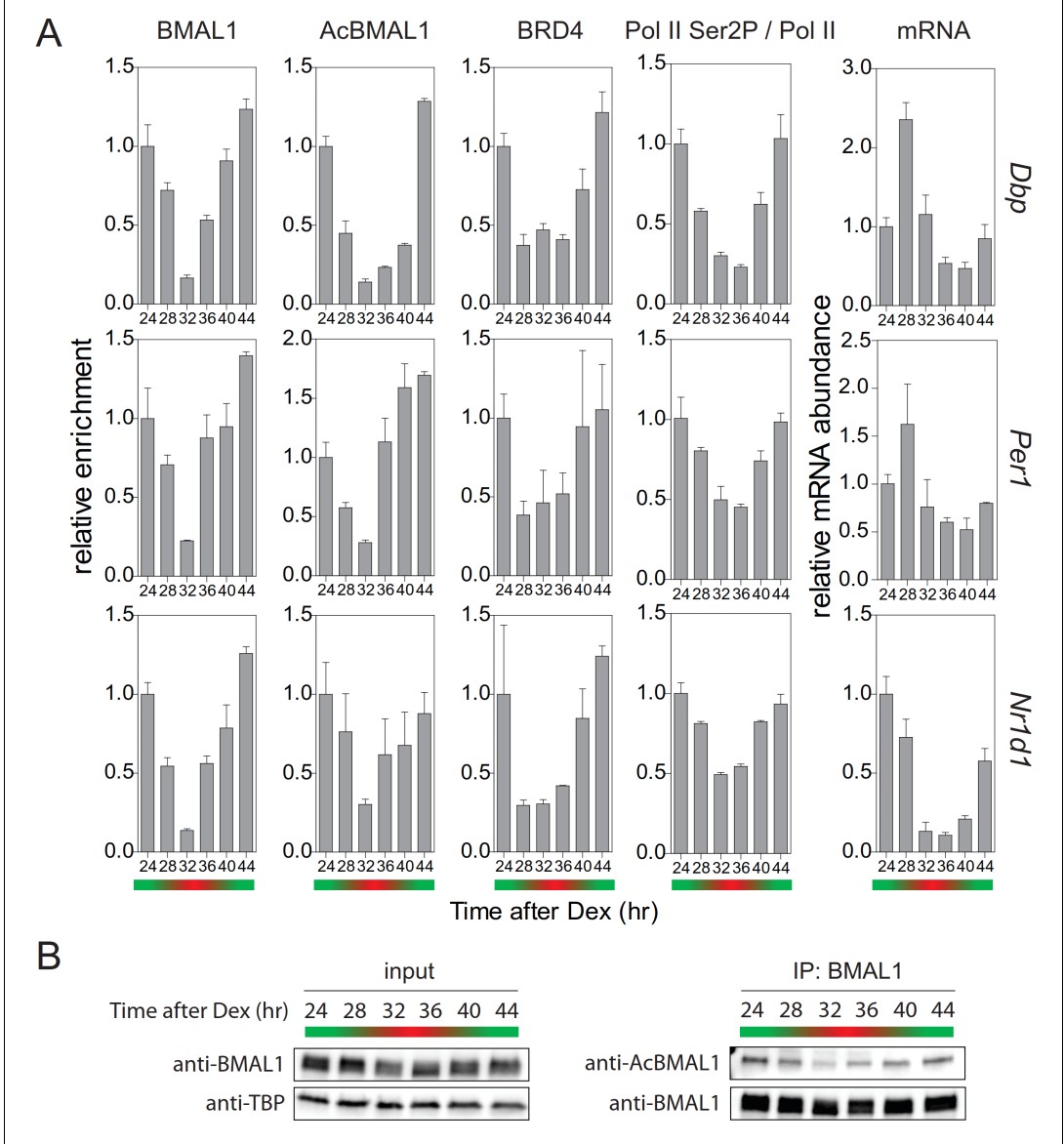

**Figure 7.** Rhythmic profile of productive elongation. (**A**) ChIP profiles of Dex-synchronized MEFs over a course of 24 hr. ChIP shows time-of-day-dependent enrichment of BMAL1, acetylated BMAL1, BRD4, and Ser2P-Pol II (normalized to total Pol II) at the promoter or 3'-end (Ser2P-Pol II) of *Dbp*, *Per1*, and *Nr1d1* genes. The rightmost column reveals time-dependence of mRNA accumulation for these genes. Data are shown as mean ± SD (n = 3). (**B**) Immunoblot analysis with the indicated antibodies of Dex-synchronized fibroblast nuclear extracts and BMAL1 IPs. The color bars represent the activation (green) and repression (red) phases of the circadian cycle.

DOI: https://doi.org/10.7554/eLife.43235.015

The following figure supplement is available for figure 7:

**Figure supplement 1.** Rhythmic profile of Pol II and Ser2P-Pol II abundance ChIP profiles of Dex-synchronized MEFs over a course of 24 hr.
DOI: https://doi.org/10.7554/eLife.43235.016

## Discussion

Genome-wide ChIP studies (*Koike et al., 2012*; *Menet et al., 2012*; *Rey et al., 2011*) show that at the onset of the circadian cycle, the circadian repressor protein CRY1 is associated with chromatin-bound CLOCK-BMAL1 located near Pol II. In this complex Pol II is bound to the clock gene promoters but is not competent to drive RNA synthesis. At this time in the circadian cycle Pol II is phosphorylated at Ser5 (*Koike et al., 2012*), which is indicative for promoter-proximal pausing and

recruitment of the mRNA capping complex to Pol II (*Harlen and Churchman, 2017*). Evidence for Pol II pausing to occur at this time at circadian promoters can also be found in global run-on experiments measuring transcriptionally engaged Pol II during the circadian cycle (*Fang et al., 2014*). Strong enrichment of Pol II at the TSS of the *Nr1d1* gene was observed at the onset of the circadian cycle, suggesting that Pol II is in a paused state. Genome-wide and circadian time-resolved quantitative analysis of Pol II abundance at TSSs and gene bodies show a peak of Pol II binding at TSSs, but not the gene bodies of clock genes, at the onset of the circadian cycle (*Zhu et al., 2018*). A circadian control of promoter proximal pausing and also of Pol II pause release would provide mechanisms capable to accurately time circadian transcription (*Jonkers and Lis, 2015*; *Liu et al., 2015*).

In the present work, we have addressed the mechanism of the circadian regulation of Pol II pause release with a focus on proteins that are known to mediate this process. We propose that pause release involves a TIP60-mediated on-chromatin acetylation of Lys538 of BMAL1. BMAL1 acetylation allows the recruitment of the co-activator BRD4 to AcBMAL1. BRD4, in turn, recruits the pause release factor P-TEFb, whose kinase subunit CDK9 then phosphorylates Ser2 of Pol II leading to a release of Pol II from the paused state enabling productive elongation of *E-box*-containing circadian genes (*Figure 7A*). In TIP60-deficient cells, Lys538 acetylation of BMAL1 was strongly diminished, BRD4-P-TEFb recruitment was reduced, and the expression of *Dbp*, *Per1*, and *Nr1d1* genes was dampened. If acetylation of BMAL at Lys538 was prevented by a Lys to Arg substitution (BMAL1$^{K538R}$), this mutant protein still interacted with TIP60, but recruitment of BRD4 and P-TEFb to the complex was reduced and Pol II pause release and productive elongation were diminished. The dysregulation of the circadian rhythm of locomotor activity in the mouse and perturbation of circadian gene expression in the SCN and in MEFs provide strong genetic evidence for a key role of TIP60 in the circadian clock. Our studies of TIP60-deficient MEFs suggest that rhythmic control of Pol II pause release may be a main process in which TIP60 is involved. Our studies thus implicate Pol II pause release as an additional regulatory step that contributes to the temporal control of the mammalian circadian clock. However, we cannot completely rule out the possibility that the loss of TIP60 could affect post-initiation event other than Pol II pause release and thereby control RNA processing or RNA stability (*Core and Adelman, 2019*). Kinetic analysis of all steps of the transcription cycle could address this issue.

TIP60 is evolutionarily highly conserved and is involved in the regulation of a series of cellular processes (see Introduction). Thereby, TIP60 has the role of an enzymatically active cofactor in larger multiprotein complexes (*Jacquet et al., 2016*; *Judes et al., 2015*). Our studies suggest a similar function of TIP60 within the circadian clock.

Our studies are based on three robustly rhythmic and clock phase-specific genes that harbor multiple CLOCK-BMAL1 binding sites such as the tandem *E-box* motif (E1-E2) (*Paquet et al., 2008*; *Rey et al., 2011*). There are many other strongly rhythmic, phase specific CLOCK-BMAL1-target genes that also harbor multiple *E-boxes* (*Rey et al., 2011*; *Shimomura et al., 2013*). This suggests that regulation by Pol II pause release could extend to the entire class of *E-box* controlled clock genes. These genes are synchronously expressed and would benefit from a precisely timed Pol II pause release process. It is not clear, however, whether BRD4-P-TEFb mediated Pol II pause release applies for other types of clock-controlled genes. Temporal analyses of Pol II pause release at the transcriptome-wide scale can address these issues.

We found that in fibroblasts, but also in the SCN, deletion of TIP60 resulted in a nearly complete loss of rhythmic expression of numerous circadian genes suggesting that in these tissues and for these genes TIP60 function is non-redundant with CBP/p300 acetyltransferase that also binds to circadian promoters. The strongest argument for BMAL1 being a substrate of TIP60 is that recombinant TIP60 directly acetylates recombinant BMAL1. Although p300 can acetylate BMAL1 it does so at lysine 500 (*Weinert et al., 2018*). The abundance of BRD4 at clock gene promoters was drastically reduced in BMAL1$^{K538R}$ cells, but it is likely that there are other acetylated substrates, such as histones H3 and H4 (*Etchegaray et al., 2003*; *Koike et al., 2012*; *Ripperger and Schibler, 2006*; *Vollmers et al., 2012*), to which BRD4 could also be recruited and affect the circadian clock.

It is proposed that CLOCK acetylates its heterodimeric partner BMAL1 at lysine 538. This leads to recruitment of CRY protein and, thus, initiates the repression phase of the circadian cycle (*Hirayama et al., 2007*). We found that TIP60-mediated acetylation of BMAL1 occurs during the activation phase of the cycle, which is consistent with the data of Nakahata and co-workers (*Nakahata et al., 2008*; *Nakahata et al., 2009*). Our finding of strong acetylation of Lys538 of

BMAL1 in CLOCK-deficient fibroblasts and the absence of an effect on the BMAL1 acetylation status in a reaction with a catalytically inactive CLOCK mutant protein raises some doubt about the efficiency of CLOCK as an acetyltransferase for BMAL1. It is noteworthy that CLOCK acetylates argininosuccinate synthase in the cytosol of U2OS cells (*Lin et al., 2017*). Since our experiments aimed at transcriptional regulation of the clock, they were carried out with nuclear extracts, they do not address cytosolic acetyltransferase activity of CLOCK.

TIP60 acetylates BMAL1 during the activation phase, but it is not clear by which mechanism BMAL1 is deacetylated during the repression phase. One possibility is a SIRT1-dependent deacetylation of BMAL1 (*Nakahata et al., 2008*). Alternatively, recruitment of histone deacetylases by PER to the CLOCK-BMAL1 heterodimer (*Duong et al., 2011*; *Kim et al., 2014*) would lead to deacetylation of BMAL1. Another possibility is that promoter-associated acetylated BMAL1 could be subject to rapid proteolytic turnover (*Stratmann et al., 2012*) obviating the need for a specific deacetylase.

Taken together, our study provides evidence that the positive limb of the circadian clock requires TIP60-mediated acetylation of BMAL1, at least for the genes studied here. This could allow a precise and synchronous expression of *E-box*-containing clock genes and in this way also contribute to robustness and precision of the circadian clock. Robust and synchronous expression are the hallmarks of genes involved in stimulus-controlled pathways, whose transcription is also often regulated through Pol II pause release (*Jonkers and Lis, 2015*; *Liu et al., 2015*; *Sawarkar et al., 2012*; *Siegal and Rushlow, 2012*).

## Materials and methods

### *Tip60* targeting vector and generation of experimental animals

Bacteria (Strain EL350) and plasmids (pL451 and pL452) used for recombineering were obtained from the NCI (Biological Resources Branch). A 15.7 kb fragment of *Tip60* genomic DNA was cloned from a BAC clone (bMQ-331N14, Sanger Institute) into a targeting vector carrying a *Pol2-DTA* cassette for negative selection. The first *loxP* site was inserted 637 bp upstream of the *Tip60* ATG. The second *loxP* site together with a *FRT*-flanked *PGK-neo* cassette was inserted 3,065 bp downstream of the ATG. The 5' end of the homologous arm of the targeting vector was 4.5 kb and the 3' end of the homologous arm was 7.6 kb. ES cell targeting and generation of $Tip60^{fl/+}$ founder mice was commissioned to PolyGene Transgenics (Switzerland). The *FRT*-flanked *PGK-neo* selection cassette was removed by crossing mice with a Flippase-expressing deleter line. Primers P1-P3 used for PCR genotyping of the Tip60 mice are listed in *Supplementary file 2*.

To generate animals for locomotor activity recording, $Tip60^{fl/fl}$ mice were crossed to a *Syt10Cre* driver mouse line resulting in Cre-mediated deletion of TIP60 predominately in the SCN (*Husse et al., 2011*). To generate $Tip60^{fl/-}$ mice for subsequent MEF isolation, $Tip60^{fl/fl}$ mice were crossed to a ubiquitous *CMVCre* driver mouse line to generate $Tip60^{+/-}$ ± which were then used for breeding with $Tip60^{fl/fl}$ mice to obtain $Tip60^{fl/-}$ offspring. For both types of studies, mice were previously backcrossed to a C57BL/6 background for at least 10 generations. Mouse handling was carried out in accordance with the German Law on Animal Welfare and was ethically approved and licensed by the Office of Consumer Protection and Food Safety of the State of Lower Saxony (license numbers 33.11.42502-04/072/07 and 33.9-42502-04-12/0719).

### Generation of $Bmal1^{K538R}$ mutant cells

$Bmal1^{K538}$ mutant cell lines were generated using CRISPR/Cas9-based genome editing. Fibroblasts stably expressing a clock-driven luciferase reporter (*Bmal1-LUC*) (*Nagoshi et al., 2004*) were co-transfected with an all-in-one plasmid pSpCas9(BB)−2A-Puro (PX459) harboring CAS9, puromycin-resistance gene, and guide RNA (sgRNA), and a single-stranded oligodeoxynucleotide (ssODNs) template comprising the point mutation. The sgRNA was designed using the online tool (http://crispr.mit.edu/). The target sequence of sgRNA was 5'-<u>GGT</u>CCTCCGTTCTTCCATTCGTA-3' (PAM sequence is underlined). Transfected cells were selected with puromycin (1 µg/ml) for two days and clonal cell lines were isolated by limiting dilution. Genomic DNA of these clones was extracted and subjected to sequencing. Primers used for screening are listed in *Supplementary file 2*.

## Plasmids

*Bmal1*, *Tip60* ORFs were amplified by PCR and cloned into *pcDNA3.1* vector containing either C-terminal myc- or V5-tags. Single site mutations were introduced by using the QuikChange II Site-directed Mutagenesis Kit (Agilent Technologies). *Tip60^{V5}* and *Tip60^{C369A;E403Q;V5}* were cloned into *pLenti PGK Hygro DEST* vector. *Bmal1-FLAG* was from M.J. Rossner (LMU Munich). *Myc-Clock* and *myc-Clock^{mutA}* were from P. Sassone-Corsi. Lentiviral *Bmal1-dLuc* reporters were from S.A. Kay (USC Los Angeles). *pBABE-puro SV40 LT* (#13970), *pCL-Eco* (#12371), *pMD2.G* (#12259), *psPAX2* (# 12260), *pLenti PGK Hygro DEST* (#19066), and *pSpCas9(BB)−2A-Puro* (# 48139) plasmids were from Addgene.

## RNA analysis by quantitative Real-Time PCR (qPCR)

Total RNA was extracted using RNeasy Plus Mini Kit (Qiagen). cDNA was synthesized using Maxima H Minus First Strand cDNA Synthesis Kit (Thermo Fisher Scientific). Real-time PCR reactions were performed on a CFX96 Real-Time PCR Detection System (Bio-Rad) with IQ SYBR Green Supermix (Bio-Rad). *Eef1a* or *Gapdh* expression was used for normalization. Primers were as described (*Oster et al., 2006*) except for *Gapdh, TIP60, 18S, Actb*, and *36b4* (*Supplementary file 2*).

## Radioactive in situ Hybridization

Mice were entrained for about 2 weeks to a 12 hr:12 hr light dark cycle and subsequently released into constant darkness. On the second day after lights-off mice were sacrificed at four time points and brains were collected and frozen in O.C.T. medium (Tissue-Tek). Templates for *Dbp, Per1*, and *Bmal1* were as described (*Oster et al., 2003*; *Oster et al., 2006*). In situ hybridization with $^{35}$S-labeled antisense RNA probes was carried out on frozen sections through the SCN. Relative quantification of expression strength was performed by densitometric analysis of autoradiograph films.

## Immunohistochemistry

Cryo-sections through the SCN were fixed with 4% paraformaldehyde for 15 min at 4°C, boiled several times in 10 mM sodium citrate (pH 6.4) for 5 min, and permeabilized for 15 min with 0.5% Triton-X-100. Sections were blocked with 10% FBS in PBS followed by incubation with anti-TIP60 antibody and a fluorescent secondary antibody. Nuclei were stained with DAPI.

## Cell culture

*Tip60^{fl/-}* MEFs were isolated following standard procedures. MEFs, fibroblasts (NIH-3T3 *Bmal1-LUC*; *Nagoshi et al., 2004*), and HEK293T cells were cultured and passaged in standard medium (DMEM, high glucose, GlutaMAX supplement, pyruvate; 10% fetal bovine serum; 100 U/ml penicillin and 100 μg/ml streptomycin [all Thermo Fisher Scientific]) at 37°C in a humidified incubator with 5% $CO_2$. MEFs and fibroblasts were synchronized by treatment with 100 nM dexamethasone, a synthetic glucocorticoid that induces expression of *Per1* (*Reddy et al., 2009*; *Reddy et al., 2012*). JQ1 (500 nM), Flavopiridol (300 nM) or vehicle (final 0.05% DMSO) were added to standard medium after synchronization. All cells were tested using morphology and PCR-based approaches to confirm their identity. Cell lines were tested negative for mycoplasma (Sigma-Aldrich).

## Transfection and viral transduction

HEK293T cells were transiently transfected using 1 mg/ml polyethylenimine solution (Sigma-Aldrich). After two days, cells were used for western blotting and immunoprecipitations (IPs). *Tip60^{fl/-}* MEFs were immortalized with SV40 LT retrovirus. Immortalized MEFs were grown to confluence and transduced with AdGFP or AdCre adenoviruses (Vector Development Laboratory, Baylor College of Medicine, Houston, TX, USA) in standard medium containing 3% fetal bovine serum (low-serum) and 8 μg/ml polybrene (Sigma-Aldrich). After two days, medium was changed to low-serum medium and cells were cultured for another 4–6 days. For lentiviral transduction (luciferase experiments) viruses were produced (*Ramezani and Hawley, 2002*) and concentrated using Lenti-X Concentrator (Clontech).

## Preparation of cell and nuclear extracts

Transiently transfected HEK293T cells were lysed in 50 mM Tris-HCl, pH 8.0; 150 mM NaCl; 5 mM EDTA; 15 mM MgCl2; 1% NP40; 1 mM DTT; 10 mM NaF; 10 mM NAM; Complete EDTA-free (Roche) for 30 min on ice and lysate was cleared by centrifugation (20,000 x g, 4°C, 10 min). Samples were boiled in SDS sample buffer or used for IP. Anti-Myc agarose beads (Sigma-Aldrich) were used to precipitate myc-tagged proteins. The precipitates were washed three times with lysis buffer and bound proteins eluted by denaturation in SDS sample buffer.

Nuclear extracts were prepared as described (Dimauro et al., 2012). Nuclei were lysed with 15 passages through an 18-gauge needle, sonicated using Bioruptor (30 s ON/30 s OFF) and cleared by centrifugation (9,000 g, 4°C, 30 min). For western blotting samples were denatured in SDS sample buffer. For IPs samples were diluted 3-fold in dilution buffer (20 mM HEPES pH 7.9; 1.5 mM MgCl2; 0.2 mM EDTA; 20% glycerol; 10 mM NAM; Complete EDTA-free [Roche]) and incubated for 2 hr at 4°C on a rotating wheel. Antibody was added and samples were put back on the rotating wheel. The next day, antibody-protein complexes were collected by Dynabead Protein G Beads (Thermo Fisher Scientific) and washed three times with wash buffer (20 mM HEPES pH 7.9; 1.5 mM MgCl$_2$; 166 mM NaCl; 0.2 mM EDTA; 20% glycerol; 0,33% Triton-X-100; 10 mM NAM; Complete EDTA-free [Roche]). Proteins were eluted by boiling in SDS sample buffer and analyzed by western blotting. For acety-lated BMAL1, western blotting was first performed with AcBMAL1 antibody followed by a harsh stripping (Abcam) step and subsequent reaction with the pan BMAL1 antibody. Signal intensities were determined on an ImageQuant LAS4010 imager (General Electric) and quantified using Image-Quant software.

## Chromatin immunoprecipitation (ChIP)

For BMAL1 (NB100-2288), acetylated BMAL1 (AB15396), BRD4 (A301-985A), CDK9 (sc-484 or sc-13130), TFIIEα (ab28177), Pol II (MABI0601), and Ser2 phosphorylated Pol II (04–1571) ChIP analysis were performed as described (Lin et al., 2012) with minor modifications. Fibroblasts were cross-linked with 1% formaldehyde for 10 min at room temperature. The reaction was stopped by adding 125 mM glycine and incubating for 5 min. Cells were washed twice with PBS, scraped and frozen in liquid nitrogen. Dynabead Protein G Beads (Thermo Fisher Scientific) were blocked with 0.5% BSA (w/v) in PBS and bound to the indicated antibodies. Crosslinked cells were lysed in lysis buffer one and washed with lysis buffer 2. Pellets were resuspended and sonicated in lysis buffer 3 (Bioruptor; 30 s ON/30 s OFF) to obtain chromatin fragments of 150–600 bp length for promoter analysis and 100–300 bp for TSS analysis to allow a higher resolution. Sonicated lysates were cleared and incu-bated overnight at 4°C with beads bound with antibody for indicated factors. Beads were washed twice (CDK9, TFIIE, and Pol II Ser2P) or three times (BMAL1, AcBMAL1, Pol II, and BRD4) with lysis buffer 3, once with lysis buffer 3 with 500 mM NaCl, one time with LiCl wash buffer and one time with TE plus 50 mM NaCl. DNA was eluted in elution buffer. The eluted complexes were de-cross-linked overnight and RNA and protein were digested using RNase A and Proteinase K. Immunopre-cipitated DNA fragments were purified by phenol chloroform extraction and ethanol precipitation and used for qPCR analysis.

For TIP60$^{V5}$ ChIP assays were performed as described (Adli and Bernstein, 2011; Nelson et al., 2006) with minor modifications. For dual cross-linking Tip60$^{-/-}$; Tip60$^{V5}$ MEFs were fixed with 1.5 mM ethylene glycol bis(succinimidylsuccinate) (EGS) (Pierce) for 30 min followed by 1% formalde-hyde for another 15 min. After quenching the reaction by adding glycine to a final concentration of 125 mM, cells were lysed in SDS lysis buffer on ice for 10 min, with brief vortexing. Samples were diluted 10-fold in ChIP dilution buffer and sonicated (Bioruptor; 30 s ON/30 s OFF) to obtain chro-matin fragments of 150–600 bp length. V5 (R96025) antibody was added to the chromatin solution and incubated overnight at 4°C on a rotating wheel. Immunoprecipitated chromatin complexes (Dynabead Protein G Beads) were washed twice with ice cold low-salt wash buffer, twice with ice cold lithium chloride wash buffer, and twice with ice cold TE buffer. 100 µl 10% Chelex 100 Slurry (Bio Rad) was added directly to the washed beads, briefly vortexed, and boiled for 10 min. Input DNA samples were ethanol precipitated and washed with 70% ethanol. DNA pellets were dissolved in 100 µl 10% Chelex 100 suspension and boiled for 10 min. Samples were Proteinase K treated and boiled for 10 min. Supernatant was transferred to a new tube and used for qPCR analysis.

Primer pair that targets an unbound region was used as internal control (*Duong et al., 2011*) and fold enrichment was calculated using the 2(-ΔΔCT) method. Experiments were performed with biological independent triplicates and data normalized to the mean value obtained from the locus with the strongest enrichment. Primer sequences for ChIP-qPCR are listed in *Supplementary file 1*.

### Bioluminescence recording

After synchronization of cells with dexamethasone medium was changed to recording medium (DMEM with high glucose, without glutamine and phenol red; 2 mM GlutaMAX; 10 mM HEPES; 5% FBS; 100 U/ml penicillin and 100 µg/ml streptomycin [all Thermo Fisher Scientific] and 0.1 mM D-luciferin [Synchem]). Petri dishes were sealed with transparent plastic film and bioluminescence was recorded in a LumiCycle luminometer (Actimetrics).

For JQ1 and flavopiridol dose dependence experiments, NIH-3T3 *Bmal1-LUC* fibroblasts were plated on 35 mm dishes and cultured for 2 days to reach confluency. Cells were synchronized and medium was replaced with the recording medium containing various concentrations of JQ1, flavopiridol or vehicle (final 0.08% DMSO). For calculation of period, luminescence intensity, and damping rate parameters the curve fitting program LumiCycle (Actimetrics) was used and data obtained from the first day were excluded from analysis because of transient luminescence changes upon the medium change.

### In vitro acetylation assay

Recombinant TIP60 (SignalChem) and BMAL1 (Abnova) proteins were mixed in acetyltransferase assay buffer (50 mM Tris-HCl, pH 8.0; 0.1 mM EDTA; 0.4 mM DTT; 25 ng/ µl BSA; 5% glycerol) in the presence or absence of 10 µM acetyl-CoA and incubated for 1 hr at 30°C. The reaction was stopped by denaturation in SDS sample buffer and proteins were analyzed by western blotting.

### Flow cytometry

Cells were stained using the SYTOX AADvanced Dead Cell Stain Kit (Thermo Fisher Scientific) according to the manufacturer's instructions and samples were analyzed using the Accuri C6 Flow Cytometer (BD). To provide a positive control cells were boiled for 3 min.

## Acknowledgements

We thank Dr. JW Lough and Dr. B Amati for antibodies, Dr. MP Antoch for *Clock*[-/-] and *Bmal1*[-/-] MEFs and Dr. SA Kay, Dr. P Sassone-Corsi and MJ Rossner for sharing vectors.

## Additional information

### Funding

| Funder | Grant reference number | Author |
| --- | --- | --- |
| Volkswagen Foundation | Lichtenberg Fellowship | Henrik Oster |
| Max-Planck-Gesellschaft | Open-access funding | Gregor Eichele |

The funders had no role in study design, data collection and interpretation, or the decision to submit the work for publication.

### Author contributions

Nikolai Petkau, Conceptualization, Data curation, Validation, Investigation, Visualization, Methodology, Writing—original draft, Writing—review and editing, Designed and supervised research, Prepared the manuscript with input from all authors; Harun Budak, Generated mouse model, Performed behavior analysis; Xunlei Zhou, Methodology, Generated mouse model; Henrik Oster, Data curation, Validation, Investigation, Methodology, Writing—original draft, Writing—review and editing; Gregor Eichele, Conceptualization, Resources, Supervision, Funding acquisition, Validation, Writing—original draft, Project administration, Writing—review and editing, Design and supervise research, Prepared the manuscript with input from all authors

## Author ORCIDs

Nikolai Petkau (iD) https://orcid.org/0000-0001-9168-3473
Harun Budak (iD) https://orcid.org/0000-0002-7371-8959
Xunlei Zhou (iD) https://orcid.org/0000-0002-9635-7068
Henrik Oster (iD) https://orcid.org/0000-0002-1414-7068
Gregor Eichele (iD) https://orcid.org/0000-0002-2863-9127

## Ethics

Animal experimentation: Mouse handling was carried out in accordance with the German Law on Animal Welfare and was ethically approved and licensed by the Office of Consumer Protection and Food Safety of the State of Lower Saxony (license numbers 33.11.42502-04/072/07 and 33.9-42502-04-12/0719).

## Decision letter and Author response

Decision letter https://doi.org/10.7554/eLife.43235.022
Author response https://doi.org/10.7554/eLife.43235.023

## Additional files

### Supplementary files

• Supplementary file 1. Two-way ANOVA statistical analysis.
DOI: https://doi.org/10.7554/eLife.43235.017

• Supplementary file 2. Primer sequences.
DOI: https://doi.org/10.7554/eLife.43235.018

• Supplementary file 3. Key resources table.
DOI: https://doi.org/10.7554/eLife.43235.019

• Transparent reporting form
DOI: https://doi.org/10.7554/eLife.43235.020

### Data availability

All data generated or analysed during this study are included in the manuscript and supporting files.

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
