## [Decision Letter]

Thank you for sending your article entitled "Acetylation of BMAL1 by TIP60 controls BRD4-P-TEFb recruitment to circadian promoters" for peer review at *eLife*. Your article is being evaluated by two peer reviewers, and the evaluation is being overseen by a Reviewing Editor and K VijayRaghavan as the Senior Editor.

The points that need to be addressed are given in the reviews below. A very important one to address is the conflict with what was previously known for BMAL1 acetylation of K538 by CLOCK leading to repression (Sassone-Corsi and co-workers) in comparison to the rather unconvincing effect of Tip60 KO on BMAL1 acetylation and interaction with Brd4 (IP-WB). Further, the way the data- analysis is presented, it is not possible to know how the ChIPs were calculated/normalized, unless this information has been missed.To reiterate, the authors would really need to address the weakness of their biochemical data. One supplementary data set arguing that CLOCK does not acetylate BMAL1 is not compelling to revise earlier reports.

*Reviewer #1:*

This interesting study presents data supporting a role of Tip60-dependant acetylation of BMAL1 to allow expression of circadian regulated genes. The acetylation of a previously identified single lysine on BMAL1 is argued to be at the center of a mechanism controlling Brd4/P-TEFb recruitment, allowing RNAPII pause release and productive transcription elongation. A large amount of data is presented to support the conclusions and endogenous acetylation site mutant of BMAL1 (K538R) is characterized after CRISPR-mediated genome editing.

While the model presented appears well supported by the data, it is very strange that it is not put in the context of known important literature directly linked to what is studied here. For example, BMAL1 acetylation on K537 (mouse, K538 in human) was published in Nature 2007 (Sassone-Corsi lab), shown to be done by CLOCK lysine acetyltransferase activity (BMAL1 heterodimer partner) and oscillates during the circadian cycle, peaking with the repression phase. It was argued that BMAL1 acetylation on K537 leads to repression of circadian regulated genes, through recruitment of CRY1. Ectopic expression of BMAL1 K537R mutant was shown to disrupt circadian rhythm. This is very different to what is proposed here.

Related to that, the authors state that they tested Tip60 as the KAT for BMAL1 because it was previously shown to co-IP with CLOCK/BMAL1 (Sassone-Corsi lab, Cell 2006). Well, that report also showed co-IP with 2 other major KATs, CBP and PCAF, all acetyltransferases previously shown to be present on E-box promoters in vivo… In fact Tip60 and PCAF histone acetyltransferase complexes are well-known co-factors (through their common TRRAP subunit) of Myc transcription factor that binds the same E-box sequences as CLOCK/BMAL1. Interestingly, elevated Myc expression was shown to disrupt the circadian clock (Cell Metabolism 2015).

Tip60/KAT5 is known to prefer the "GGK" sequence as substrate and this is what is found for K538 (BMAL1 sequence "GGKKI"). So modification in vitro with recombinant proteins (or through joint over-expression in transient transfection) makes sense but does not prove per se in vivo physiological relevance (see below).

Specific points about the data:

– Figure 2A:

The effect of K538R on co-IP of CDK9/Brd4 is difficult to judge since there seems to be less BMAL1 signal in the input and the BMAL1 WB signal is saturated. I suspect the effect is, in fact, minor if any.

– Figure 3—figure supplement 1:

This is a key experiment in the context of the literature that showed CLOCK-dependent acetylation of BMAL1. The data presented here raise some questions as there seems to be higher and more constant acBMAL1 signal in the CLOCK KO cells. From the band patterns on the CLOCK KO membrane it seems that the same membrane was used for both total BMAL1 and acBMAL1. Is it possible that the acBMAL1 signal is, in fact, residual total BMAL1 signal after stripping the membrane? Or non-specific acBMAL1 signal because of the high amount of IPed BMAL1 on the membrane?

– Figure 4C:

The effect of Tip60 induced KO on BMAL1 acetylation in vivo is not convincing since the level of total BMAL1 IPed is lower in the KO sample. The graph below based on n=3 shows a decrease after normalization but since the shown WB is not convincing I do not know what to think. Quantification of single point WB signals is known to be misleading as ECL is not linear. The decrease of acBMAL1 seems similar to the one shown for total BMAL1.

– Figure 5A:

The exact same thing can be said here as for Figures 2A and 4C. The effect of Tip60 KO on BMAL1 acetylation/interaction with Brd4 is not convincing as there is less BMAL1 in the input and the IPed signal is saturated but still points to less IPed BMAL1 in Tip60 KO). An identical problem can be noted in 5C as much less Brd4 is IPed, making the conclusion on BMAL1 interaction being dependent on Tip60 not valid.

In any case, it is clear that BMAL1 still gets acetylated in the absence of Tip60? As the TIP60 coactivator complex is present on the enhancers and promoters of a large number of transcribed genes, the effect of its induced KO can lead to indirect effects on gene transcription.

– Figure 5D:

To distinguish effects on txn initiation vs elongation the authors perform ChIP-qPCR with RNAPII-Ser2ph and TFIIE Abs. To really argue about initiation vs elongation, or pause release, the real readout should be RNAPII-Ser5ph vs Ser2ph?

Overall, all ChIP-qPCR experiments are difficult to efficiently judge as it is not clear from the Materials and methods/Figure legends how these values were calculated. 6B acBMAL1 oscillation not convincing with load. The graphs indicate "relative enrichment" and the Materials and methods state "to unbound regions". But since the values are 1 and below this would mean no enrichment vs the control unbound regions? The ChIP-qPCR methods also state n=3, is this biological independent triplicate experiments? Or technical replicates (same chromatin, 3 IPs)? Or PCR replicates?

*Reviewer #2:*

This is an interesting paper that explores mechanisms that contribute to circadian regulation of clock gene transcription, using experiments performed in cultured cells and mutant mice

In the first section of the paper, the authors show that periodic expression of a BMAL1-driven luciferase reporter and of several endogenous clock-regulated genes is inhibited by the Cdk9 inhibitor flavopiridol and by JQ1, a BET inhibitor that preferentially targets BRD4, which contributes to P-TEFb recruitment to genes. Consistent with these observations, JQ1 treatment also leads to loss of BRD4 and Cdk9 at the clock gene Dbp and to a decreased in phosphorylation of the Pol II CTD on Ser2. Based on these observations, the authors propose that BRD4-dependent P-TEFb recruitment is a rate limiting step in Dbp regulation. In further experiments, they present evidence that (i) BRD4 (and Cdk9) bind BMAL1 acetylated at lysine 538; (ii) Tip60 acetylates BMAL1 at K538; (iii) occupancy of BRD4, Cdk9, and Ser2P-phosphorylated Pol II at clock genes is reduced in cells expressing BMAL K538R, as is expression of Dbp mRNA and a BMAL1 luciferase reporter; (iv) Tip60 deficiency gives rise to a circadian phenotype in mice, dampening or disruption of cyclic clock gene expression in SCN and in fibroblasts, (v) Tip60 deficiency also leads to decreased BMAL1 acetylation, BRD4, Cdk9, and Ser2P-phosphorylated Pol II at clock genes.

Overall, the data are consistent with the authors' model that the histone acetyl transferase TIP60 contributes to circadian gene regulation by acetylating BMAL1, which in turn leads to recruitment of BRD4 and Cdk9/P-TEFb and release of promoter proximally paused Pol II into productive elongation of several circadian transcripts. I do, however, have a number of minor comments.

Comments:

1) Relevant to Figure 1 and supplemental figures: It would perhaps be more surprising if blocking P-TEFb/Cdk9 or BRD4 activity didn't interfere with clock gene regulation than that it does since they have very widespread roles in gene regulation.

2) The authors used the ratio of Ser2P Pol II/total Pol II as a measure of release from promoter-proximal pausing into productive elongation, but they don't show effects of drug treatments or BMAL1 / Tip60 mutations on total Pol II distribution. I think it would be better to show not only the ratio of these two ChIP signals but also each of them individually so the reader can assess the degree to which the mutations alter Pol II distribution vs Pol II CTD phosphorylation.

3) The authors conclude that BMAL or Tip60 mutation doesn't affect initiation based on evidence that the mutations don't lead to changes in TFIIE occupancy measured by ChIP. Strictly speaking, this is consistent with idea that PIC assembly is unlikely to be affected but doesn't rule out changes in initiation rate.

---

## [Author Response]

Reviewer #1:

[…] While the model presented appears well supported by the data, it is very strange that it is not put in the context of known important literature directly linked to what is studied here. For example, BMAL1 acetylation on K537 (mouse, K538 in human) was published in Nature 2007 (Sassone-Corsi lab), shown to be done by CLOCK lysine acetyltransferase activity (BMAL1 heterodimer partner) and oscillates during the circadian cycle, peaking with the repression phase. It was argued that BMAL1 acetylation on K537 leads to repression of circadian regulated genes, through recruitment of CRY1. Ectopic expression of BMAL1 K537R mutant was shown to disrupt circadian rhythm. This is very different to what is proposed here.

We are aware of the proposed model of Sassone-Corsi and co-workers that rests on their finding that acetylation of Lys538 by CLOCK peaks within the repression phase. This enables recruitment of CRY expressed in the repression phase. Specifically, Hirayama et al., 2007, show in their Figure 1A that BMAL1 acetylation peaks at Zeitgeber time 15 (ZT15), which is in the repression phase. However, please note that in a subsequent publications from the Sassone-Corsi laboratory (Nakahata et al., 2008), BMAL1 acetylation is most pronounced during the activation phase (left panel of Figure 1B).

These data are for the liver, the same is also true for fibroblasts. Nakahata et al., 2008, show that BMAL1 acetylation peaks ~18 h post serum shock (Figure 2A) which is the time when the BMAL1 direct target gene Dbp (for a discussion about direct targets see the Schibler lab paper by Stratmann et al., 2008) shows peak expression (Figure 2B). Accordingly, BMAL1 acetylation peaks during the activation phase at which time CRYs are of low abundance.

Nakahata et al., 2009, show that maximal BMAL1 acetylation levels again coincide with peak expression of Dbp (Figure 3).

To “put [our approach] in the context of known important literature” is rather difficult because the data from the Sassone-Corsi lab is not internally consistent. Note, however, that their 2008 and 2009 publications are consistent with our finding that maximal BMAL1 acetylation falls into the activation phase of the clock cycle.

In the Introduction we address the Sassone-Corsi results (Hiarayama et al., 2007) in the context of reviewing the ChIP-seq data of Takahashi’s group (paragraph three). We also comment on the apparent contradictions of the different papers published by the Sassone-Corsi lab (Hiarayama et al., 2007 vs. Nakahata et al., 2008/9). We come back to the point of when BMAL1 is acetylated in the Discussion.

*Related to that, the authors state that they tested Tip60 as the KAT for BMAL1 because it was previously shown to co-IP with CLOCK/BMAL1 (Sassone-Corsi lab, Cell 2006). Well, that report also showed co-IP with 2 other major KATs, CBP and PCAF, all acetyltransferases previously shown to be present on E-box promoters* in vivo.

We focus on TIP60 because knocking it out in the SCN leads to a severe circadian phenotype resembling that seen in BMAL1 deficiency. Of course, it is possible that CBP/p300 or PCAF acetylate BMAL1 at Lys538. Note, however, that CBP/p300 acetylates Lys500 of BMAL1 (Weinert et al., 2018). At the present time there is no knock out model that suggests that CBP/p300 or PCAF play a key role in circadian clock function.

We refer to the Weinert paper in the Discussion.

In fact Tip60 and PCAF histone acetyltransferase complexes are well-known co-factors (through their common TRRAP subunit) of Myc transcription factor that binds the same E-box sequences as CLOCK/BMAL1. Interestingly, elevated Myc expression was shown to disrupt the circadian clock (Cell Metabolism 2015).

If we understand the reviewer correctly, the reviewer suggests that TIP60-deficency would result in dysregulation of *Myc* expression and this would disrupt the circadian clock. Overexpression of MYC leads to enhanced expression of clock controlled genes (Altman et al., Cell Metabolism 2015; 22(6): 1009-19), a phenotype that we do not observe in TIP60-deficient cells (see Figure 3D and 3F).

*Tip60/KAT5 is known to prefer the "GGK" sequence as substrate and this is what is found for K538 (BMAL1 sequence "GGKKI"). So modification* in vitro *with recombinant proteins (or through joint over-expression in transient transfection) makes sense but does not prove per se* in vivo *physiological relevance (see below).*

Correct, *in-vitro* assays and overexpression in transient transfection experiments are supportive “only”, but we also showed that TIP60 deficiency in MEFs (derived from floxed mouse embryos) exhibited markedly reduced BMAL1 acetylation and diminished recruitment of BRD4 and p-TEFb to BMAL1 with the effect of preventing pause release.

Specific points about the data:– Figure 2A:The effect of K538R on co-IP of CDK9/Brd4 is difficult to judge since there seems to be less BMAL1 signal in the input and the BMAL1 WB signal is saturated. I suspect the effect is, in fact, minor if any.

Detection is based on chemiluminescence and measured with a GE ImageQuant LAS 4010 instrument that yields a quantitative output. Unlike X-ray films the instrument gives a linear correlation between protein quantity and signal intensity over 4 orders of magnitude. We measured the BMAL1 input signals and observed only a minor change (1.00 vs. 0.87). In a revised manuscript, we will provide numerical values next to the WB bands (see e.g. Zhang et al*., eLife* 2017; 6: e24466).

We provide numerical values for signal intensity next to the WB bands.

– Figure 3—figure supplement 1:This is a key experiment in the context of the literature that showed CLOCK-dependent acetylation of BMAL1. The data presented here raise some questions as there seems to be higher and more constant acBMAL1 signal in the CLOCK KO cells. From the band patterns on the CLOCK KO membrane it seems that the same membrane was used for both total BMAL1 and acBMAL1. Is it possible that the acBMAL1 signal is, in fact, residual total BMAL1 signal after stripping the membrane? Or non-specific acBMAL1 signal because of the high amount of IPed BMAL1 on the membrane?

Yes, the same membrane was used for total BMAL1 and acetylated BMAL1. Unlike the referee suspected, we first detected acetylated BMAL1 followed by a stripping step and reaction with the pan BMAL1 antibody. We will add this information to “Materials and methods”.

The CLOCK-deficient cells were a gift from M.P. Antoch but were re-genotyped using primers as described in Debruyne et al., 2006. These cells shown in Figure 3—figure supplement 1 are uncontestable CLOCK-deficient.

One could argue that the Millipore antibody we used is not specific for acetylated BMAL1. However, the strong specificity of this reagent is demonstrated in Figure 2—figure supplement 1D where we detect K538 acetylated BMAL1 only in cells expressing wild type BMAL1.

We propose to include a second line of evidence, a co-transfection experiment, that further tests the hypothesis that BMAL1 is a CLOCK substrate. The experiment goes as follows. BMAL1 is transfected into HEK293 cells and after 2 days of culture the extent of acetylation of BMAL1 is determined by quantitative chemiluminescence. This defines base-line Lys538 acetylation levels of BMAL1. In parallel, BMAL1 and CLOCK or CLOCK-mutA are co-transfected (CLOCK-mutA is a catalytically inactive CLOCK that was obtained from the Sassone-Corsi lab and was re-sequenced by us for verification). If CLOCK acetylates BMAL1 the level of acetylated BMAL1 should increase, while CLOCK-mutA should not have this effect. Lastly, we will co-transfect TIP60 with BMAL1 and CLOCK or CLOCK-mutA. Here we expect that TIP60 will markedly increases BMAL1 acetylation.

We note in Materials and methods that we first detected acetylated BMAL1, subsequently carried out a stripping step followed by an incubation with the pan BMAL1 antibody.

BMAL1 is transfected into HEK293 cells and after 2 days of culture the extent of acetylation of BMAL1 was determined by quantitative chemiluminescence (new Figure 3B). This defines base-line Lys538 acetylation levels of BMAL1. In parallel, BMAL1 and CLOCK or CLOCK-mutA were co-transfected. We found that both clock variants showed a 2-fold increase of BMAL1 acetylation. By contrast with TIP60 this increase was 8-fold.

– Figure 4C:The effect of Tip60 induced KO on BMAL1 acetylation in vivo is not convincing since the level of total BMAL1 IPed is lower in the KO sample. The graph below based on n=3 shows a decrease after normalization but since the shown WB is not convincing I do not know what to think. Quantification of single point WB signals is known to be misleading as ECL is not linear. The decrease of acBMAL1 seems similar to the one shown for total BMAL1.

A dilution series would address this critique. In fact, Figure 4C where we show the dependency of BRD4 recruitment on the acetylation status of BMAL1, provides such a dilution series and thus confirms the marked reduction of BMAL1 acetylation upon AdCre-mediated TIP60 deletion. We propose that in a revision, we will generate a dilution series with 3 additional biological replicates and revise Figure 4C accordingly. At the same time we will also measure TIP60 levels so as to address critique 9 below.

The new data (Figure 5C) are based on a dilution series and a small experimental modification in that compared to our previous experiment we used 25% more AdCre s and an incubation period of 6 instead of 4 days.

– Figure 5A:The exact same thing can be said here as for Figures 2A and 4C. The effect of Tip60 KO on BMAL1 acetylation/interaction with Brd4 is not convincing as there is less BMAL1 in the input and the IPed signal is saturated but still points to less IPed BMAL1 in Tip60 KO). An identical problem can be noted in 5C as much less Brd4 is IPed, making the conclusion on BMAL1 interaction being dependent on Tip60 not valid.

Please see our reply to critique point 5. We will provide numerical values demonstrating that protein loads were comparable between samples. We will repeat the experiment of Figure 5B with more similar sample concentrations.

The experiment was repeated so that the experimental and control lanes now show a similar signal strength that was quantified.

In any case, it is clear that BMAL1 still gets acetylated in the absence of Tip60?

Yes, in the absence of TIP60, BMAL1 acetylation is reduced to about 35%. It should be recalled how this experiment is carried out. First, fibroblasts are grown to confluence at which time they stop proliferation. AdCre or AdGFP is added which will then lead to the excision of the floxed TIP60 with the first vector. Eventually, TIP60 protein will also disappear. At day 3 cells are synchronized by dexamethasone treatment and lysed 24 h later. Then the acetylation status of BMAL1 is checked. This is a standard protocol, but it does not “remove” all TIP60 protein. In fact, Figure 5A shows that with this protocol there is residual TIP60 (18%). We will address the issue of residual TIP60 in the context of Critique 7 above where we quantify TIP60.

See our reply above. The reduction of BMAL1 acetylation was > than 85% in three independent biological replicate experiments.

As the TIP60 coactivator complex is present on the enhancers and promoters of a large number of transcribed genes, the effect of its induced KO can lead to indirect effects on gene transcription.

TIP60 is an enzyme and as such TIP60 has multiple substrates that play a role in diverse cellular functions. This is reflected in the early embryonic lethality of *Tip60*-deficient flies or mice. The challenge we were facing was to set up experiments in which we could study a specific TIP60 function. We did this by studying postmitotic cells (neurons of the SCN [using the CRE-LoxP system] and growth-arrested fibroblasts) and we focused on a single substrate – BMAL1 – that is part of the circadian pacemaker. Our “bias” for BMAL1 and the clock was due to the fact, that our genetic experiments revealed a strong impairment of the circadian clock (and nothing else). Hence it makes sense to look for substrates of TIP60 among the clock proteins. Three major additional pieces of information prompted us to follow this path. First, Sassone-Corsi and associates showed that BMAL1 was acetylated and that such acetylation was functionally important. Second, we had some doubts about the possibility that CLOCK acetylates BMAL1 at lysine 538, since in CLOCK-deficient fibroblasts that particular lysine was still acetylated. Nonetheless, even in this situation there could be additional TIP60 substrates that contribute to the circadian phenotype such as hypoacetylation of certain histones. We comment on this particular point in our Discussion (paragraph five).Third, in growth-arrested Tip60-deficient MEFs we found no change in transcription of *Gapdh, 18S, Actb*, and *36b4* genes Figure 3 (now Figure 4F and Figure 4 —figure supplement 1D). This would argue against a pleiotropic effect of TIP60.

– Figure 5D:To distinguish effects on txn initiation vs elongation the authors perform ChIP-qPCR with RNAPII-Ser2ph and TFIIE Abs. To really argue about initiation vs elongation, or pause release, the real readout should be RNAPII-Ser5ph vs Ser2ph?

Characterization of initiation and elongation require complex multi-omics approaches that are not just done by compering Ser2 and Ser5 phosphorylation of CTD (Gressel et al*., eLife* 2017; 6: e29736). Nevertheless, in this manuscript we clearly show that TIP60-mediated acetylation of BMAL1 regulates Pol II pause release and that preinitiation complex (PIC) assembly is not affected by TIP60 deficiency. Our paper pioneers the idea that TIP60-mediated control of BRD4-P-TEFb recruitment is a novel temporal checkpoint in the circadian cycle. Obviously, our finding will require further mechanistic analyses that investigate the complex interplay of transcription initiation and elongation.

Overall, all ChIP-qPCR experiments are difficult to efficiently judge as it is not clear from the Materials and methods/Figure legends how these values were calculated.6B acBMAL1 oscillation not convincing with load… The graphs indicate "relative enrichment" and the Materials and methods state "to unbound regions". But since the values are 1 and below this would mean no enrichment vs the control unbound regions? The ChIP-qPCR methods also state n=3, is this biological independent triplicate experiments? Or technical replicates (same chromatin, 3 IPs)? Or PCR replicates?

We apologize that the way the data analysis was presented, did not include a description of how the ChIP experiments were calculated and normalized. Biological independent triplicate experiments were performed and data were first normalized to an internal control and maximum values were set to 1. We missed to give this information in “Materials and methods” section and the requested information is now found in the Materials and method section.

Reviewer #2:

[…] Overall, the data are consistent with the authors' model that the histone acetyl transferase TIP60 contributes to circadian gene regulation by acetylating BMAL1, which in turn leads to recruitment of BRD4 and Cdk9/P-TEFb and release of promoter proximally paused Pol II into productive elongation of several circadian transcripts. I do, however, have a number of minor comments.Comments:1) Relevant to Figure 1 and supplemental figures: It would perhaps be more surprising if blocking P-TEFb/Cdk9 or BRD4 activity didn't interfere with clock gene regulation than that it does since they have very widespread roles in gene regulation.

JQ1 and Flavopiridol affect the expression of many genes, but rather than inhibiting transcription, they prevent Pol II pause release. Thus, the effect of these pharmacological agents on the clock cycle and amplitude suggests that Pol II pause and pause release are likely to be regulatory steps in the circadian clock.

A brief assessment of the usefulness and meaning of the inhibitor experiment is provided in the first paragraph of the Results section.

2) The authors used the ratio of Ser2P Pol II/total Pol II as a measure of release from promoter-proximal pausing into productive elongation, but they don't show effects of drug treatments or BMAL1 / Tip60 mutations on total Pol II distribution. I think it would be better to show not only the ratio of these two ChIP signals but also each of them individually so the reader can assess the degree to which the mutations alter Pol II distribution vs Pol II CTD phosphorylation.

We have these data and we will include them in the revision in Figures 2 and 5 and in the associated supplemental figures.

These data are include in the revision in the supplement figures of Figures 1, 2, 6, and 7.

3) The authors conclude that BMAL or Tip60 mutation doesn't affect initiation based on evidence that the mutations don't lead to changes in TFIIE occupancy measured by ChIP. Strictly speaking, this is consistent with idea that PIC assembly is unlikely to be affected but doesn't rule out changes in initiation rate.

Yes, recent studies from the Cramer and Zeitlinger laboratories show that promoter-proximal pausing of Pol II sets a limit to the frequency of transcription initiation in both human and *Drosophila* cells (Gressel et al.*, eLife* 2017; 6: e29736; Shao and Zeitlinger, Nat. Genet. 2017; 49: 1045–1051). Pol II pausing inhibits new transcription initiation. Therefore, recruitment of P-TEFb to the promoters, as described in our study, plays a central role in controlling pause duration and thereby also the productive initiation frequency (Gressel et al., *eLife* 2017; 6: e29736).